# The Role of Permutation Invariance in Linear Mode Connectivity of Neural Networks

Rahim Entezari[1], Hanie Seghi[2], Olga Saukh[1], and Behnam Neyshabur[3]

[1]TU Graz / CSH Vienna, [2]Google Research, Brain Team, [3]Google Research, Blueshift Team

## ABSTRACT

In this paper, we conjecture that if the permutation invariance of neural networks is taken into account, SGD solutions will likely have no barrier in the linear interpolation between them. Although it is a bold conjecture, we show how extensive empirical attempts fall short of refuting it. We further provide a preliminary theoretical result to support our conjecture. Our conjecture has implications for lottery ticket hypothesis, distributed training and ensemble methods. The source code is available at `https://github.com/rahimentezari/PermutationInvariance`.

## 1 INTRODUCTION

Understanding the loss landscape of deep neural networks has been the subject of many studies due to its close connections to optimization and generalization (Li et al., 2017; Mei et al., 2018; Geiger et al., 2019; Nguyen et al., 2018; Fort et al., 2019; Baldassi et al., 2020). Empirical observations suggest that loss landscape of deep networks has many minima (Keskar et al., 2017; Draxler et al., 2018; Zhang et al., 2017). One reason behind the abundance of minima is over-parametrization. Over-parametrized networks have enough capacity to present different functions that behave similarly on the training data but vastly different on other inputs (Neyshabur et al., 2017; Nguyen et al., 2018; Li et al., 2018; Liu et al., 2020). Another contributing factor is the existence of scale and permutation invariances which allows the same function to be represented with many different parameter values of the same network and imposes a counter-intuitive geometry on the loss landscape (Neyshabur et al., 2015; Brea et al., 2019a).

Previous work study the relationship between different minima found by SGD and establish that they are connected by a path of non-increasing loss; however, they are *not* connected by a *linear* path (Freeman & Bruna, 2016; Draxler et al., 2018; Garipov et al., 2018). This phenomenon is often referred to as mode connectivity (Garipov et al., 2018) and the loss increase on the path between two solutions is often referred to as (energy) barrier (Draxler et al., 2018). Understanding *linear mode connectivity* (LMC) is highly motivated by several direct conceptual and practical implications from pruning and sparse training to distributed optimization and ensemble methods.

The relationship between LMC and pruning was established by Frankle et al. (2020) where they showed the correspondence between LMC and the well-known lottery ticket hypothesis (LTH) (Frankle & Carbin, 2019). In short, LTH conjectures that neural networks contain sparse subnetworks that can be trained in isolation, from initialization, or early in training to achieve comparable test accuracy. Frankle et al. (2020) showed that solutions that are linearly connected with no barrier have the same lottery ticket. They further discuss how linear-connectivity is associated with stability of SGD. This view suggests that SGD solutions that are linearly connected with no barrier can be thought of as being in the same basin of the loss landscape and once SGD converges to a basin, it shows a stable behavior inside the basin[1]. Because of the direct correspondence between LMC and LTH, any understanding of LMC, has implications for LTH, stability of SGD and pruning techniques.

Linear mode connectivity has also direct implications for ensemble methods and distributed training. Ensemble methods highly depend on an understanding of the loss landscape and being able to sample from solutions. Better understanding of mode connectivity has been shown to be essential in devising better ensemble methods (Garipov et al., 2018). Linear mode connectivity between solutions or checkpoints also allows for weight averaging techniques for distributed optimization to be used as effectively in deep learning as convex optimization (Scaman et al., 2019).

---

[1]This notion of basin is also consistent with the definition proposed by Neyshabur et al. (2020).

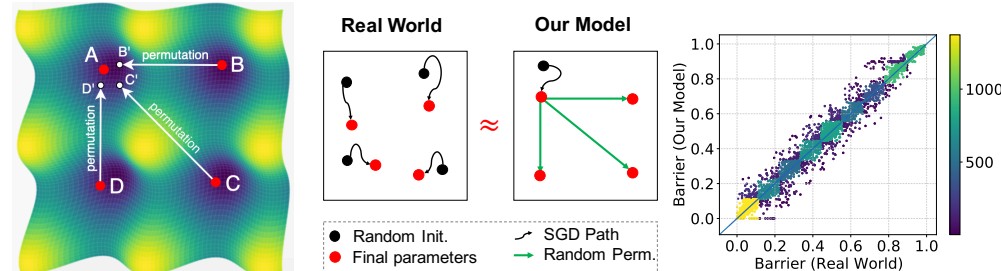

Figure 1: **Linear mode connectivity when using permutation invariance**. **Left:** Schematic picture of four minima $A, B, C, D$ in different basins with an energy barrier between each pair. However, our conjecture suggests that permuting hidden units of $B, C$ and $D$ would result in $B', C'$ and $D'$ which present the exact same function as before permutation while having no barrier on their linear interpolation with $A$. **Middle:** Our model for *barriers* in real world SGD solutions. In real world we train networks by running SGD with different random seeds starting from different initializations. In our model, different final networks are achieved by applying random permutations to the same SGD solution (or equivalently, applying random permutations to the same initialization and then running SGD with the same seed on them). **Right:** Aggregation of our extensive empirical evidence (more than 3000 trained networks) in one density plot comparing barriers in real world against our model across different choices of architecture family, dataset, width, depth, and random seed. Points in the lower left mostly correspond to lowest barrier found after searching in the space of valid permutations using a Simulated Annealing (SA). For a detailed view on architectures and datasets see Figure 10 in Appendix A.3.

In this paper, we conjecture that by taking permutation invariance into account, the loss landscape can be simplified significantly resulting in linear mode connectivity between SGD solutions. We investigate this conjecture both theoretically and empirically through extensive experiments. We show how our attempts fall short of refuting this hypothesis and end up as supporting evidence for it (see Figure 1). We believe our conjecture sheds light into the structure of loss landscape and could lead to practical implications for the aforementioned areas.

**Contributions**. This paper makes the following contributions:

- We study linear mode connectivity (LMC) between solutions trained from different initializations and investigate how it is affected by choices such as width, depth and task difficulty for fully connected and convolutional networks ( Section 2).

- We introduce our main conjecture in Section 3: If invariances are taken into account, there will likely be no barrier on the linear interpolation of SGD solutions (see the left panel of Figure 1).

- By investigating the conjecture theoretically, we prove that it holds for a wide enough fully-connected network with one hidden layer at random initialization ( Section 3).

- In Section 4, we provide strong empirical evidence in support of our conjecture. To overcome the computational challenge of directly evaluating the hypothesis empirically, which requires searching in the space of all possible permutations, we propose an alternative approach. We consider a set of solutions corresponding to random permutations of a single fixed SGD solution (our model) and show several empirical evidences suggesting our model is a good approximation for all SGD solutions(real world) with different random seeds (see the middle and right panel of Figure 1).

**Further related work**. Permutation symmetry of neurons in every layer results in multiple equivalent minima connected via saddle points. Few studies investigate the role of these symmetries in the context of connectivity of different basins. Given a network with L layers of minimal widths $r_1^*, ..., r_{L-1}^*$ that reaches zero-loss minima at $r_1!, ..., r_{L-1}!$ isolated points (permutations of one another), Şimşek et al. (2021) showed that adding one extra neuron to each layer is sufficient to connect all these previously discrete minima into a single manifold. Fukumizu & Amari (2000) prove that a point corresponding to the global minimum of a smaller model can be a local minimum or a saddle point of the larger model. Brea et al. (2019b) find smooth paths between equivalent global minima that lead through a permutation point, *i.e.*, where the input and output weight vectors of two neurons in the same hidden layer interchange. They describe a method to permute all neuron indices in the same layer at the same cost. Singh & Jaggi (2020) proposed a layer-wise model fusion algorithm for making ensembles. Their method utilizes optimal transport for aligning neurons across the models trained from different initializations. Tatro et al. (2020) showed that aligning the neurons

in two different neural networks makes it easier to find second order curves between them in the loss landscape where barriers are absent.

## 2 LOSS BARRIERS

In this section, we first give a formal definition for linear mode connectivity and study how it is affected by different factors such as network width, depth, and task difficulty for a variety of architectures.

### 2.1 DEFINITIONS

Let $f_\theta(\cdot)$ be a function presented by a neural network with parameter vector $\theta$ that includes all parameters and $\mathcal{L}(\theta)$ be the any given loss (e.g., train or test error) of $f_\theta(\cdot)$. Let $\mathcal{E}_\alpha(\theta_1, \theta_2) = \mathcal{L}(\alpha\theta_1 + (1-\alpha)\theta_2)$, for $\alpha \in [0, 1]$ be the loss of the network created by linearly interpolating between parameters of two networks $f_{\theta_1}(\cdot)$ and $f_{\theta_2}(\cdot)$. The loss barrier $B(\theta_1, \theta_2)$ along the linear path between $\theta_1$ and $\theta_2$ is defined as the highest difference between the loss occurred when linearly connecting two points $\theta_1, \theta_2$ and linear interpolation of the loss values at each of them:

$$B(\theta_1, \theta_2) = \sup_\alpha [[\mathcal{L}(\alpha\theta_1 + (1-\alpha)\theta_2)] - [\alpha\mathcal{L}(\theta_1) + (1-\alpha)\mathcal{L}(\theta_2)]]. \tag{1}$$

The above definition differs from what was proposed by Frankle et al. (2020) in that they used $0.5\mathcal{L}(\theta_1) + 0.5\mathcal{L}(\theta_2)$ instead of $\alpha\mathcal{L}(\theta_1) + (1-\alpha)\mathcal{L}(\theta_2)$ in our definition. These definitions are the same if $\mathcal{L}(\theta_1) = \mathcal{L}(\theta_2)$. But if $\mathcal{L}(\theta_1), \mathcal{L}(\theta_2)$ are different, we find our definition to be more appropriate because it assigns no barrier value to a loss that is changing linearly between $\theta_1$ and $\theta_2$. We say that two networks $\theta_1$ and $\theta_2$ are linear mode connected if the barrier between them along a linear path is $\approx 0$ (Frankle et al., 2020). It has been observed in the literature that any two minimizers of a deep network can be connected via a non-linear low-loss path (Garipov et al., 2018; Draxler et al., 2018; Fort & Jastrzebski, 2019). This work examines *linear* mode connectivity (LMC) between minima. Next, we empirically investigate the effect of task difficulty and choices such as architecture family, width and depth on LMC of SGD solutions.

### 2.2 EMPIRICAL INVESTIGATION: BARRIERS

In this section, we look into barriers between different SGD solutions on all combinations of four architecture families (MLP (Rosenblatt, 1961), Shallow CNN (Neyshabur, 2020), ResNet (He et al., 2015) and VGG (Simonyan & Zisserman, 2015)) and four datasets (MNIST (LeCun & Cortes, 2010), SVHN (Netzer et al., 2011), CIFAR-10 (Krizhevsky et al., 2009) and CIFAR-100 (Krizhevsky et al., 2009)). The main motivation to use Shallow CNN is to move from fully connected layers (MLP) to convolutions. The main difference between Shallow CNN and VGG16 is depth and the main difference between ResNet18 and VGG16 is existence of residual connections. We empirically investigate how different factors such as architecture family, width, depth and task difficulty impact the barrier size[2]. We refer to training loss barrier as barrier. For loss barriers on a test set see E.4. For train and test errors see A.2 .

**Width:** We evaluate the impact of width on the barrier size in Figure 2. We note that for large values of width the barrier becomes small. This effect starts at lower width for simpler datasets such as MNIST and SVHN compared to CIFAR datasets. A closer look reveals that the barrier increases with width up to a point and beyond that increasing width leads to lower barrier size. This effect is reminiscent of the double descent phenomena (Belkin et al., 2019; Nakkiran et al., 2019). Checking the test error (Figure 8) indicates that in our experiments the barrier peak happens at the same size that needed to fit the training data. This phenomena is observed for both fully-connected

---

[2]In all plots the barrier is evaluated using training loss across 5 different random pairs (10 random SGD solutions). For easier comparison between all figures, we report the train accuracy barrier in all barrier plots. In our experiments, we observed that evaluating the barrier at $\alpha = \frac{1}{2}$ is a reasonable surrogate for taking the supremum over $\alpha$ (the difference is less than $10^{-4}$). Therefore, to save computation, we report the barrier value at $\alpha = \frac{1}{2}$.

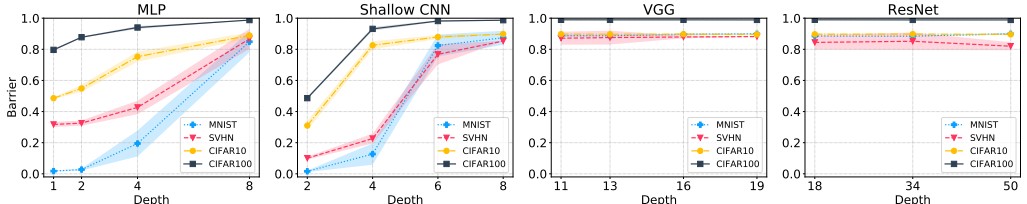

Figure 2: **Effect of width on barrier size.** From **left to right**: one-layer MLP, two-layer Shallow CNN, VGG-16 and ResNet-18 architectures on MNIST, CIFAR-10, SVHN, CIFAR-100 datasets. For large width sizes the barrier becomes small. This effect starts at lower width for simpler datasets such as MNIST and SVHN compared to CIFAR datasets. A closer look reveals a similar trend to that of double-descent phenomena. MLP architectures hit their peak at a lower width compared to CNNs and a decreasing trend starts earlier. For ResNet, the barrier size is saturated at a high value and does not change due to the effect of depth as discussed in Figure 3.

Figure 3: **Effect of depth on barrier size.** From **left to right** MLP, Shallow CNN, VGG(11,13,16,19), and ResNet(18,34,50) architectures on MNIST, CIFAR-10, SVHN, CIFAR-100 datasets. For MLP and Shallow CNN, we fix the layer width at $2^{10}$ while adding identical layers as shown along the x-axis. Similar behavior is observed for fully-connected and CNN family, *i.e.*, low barrier when number of layers are low while we observe a fast and significant barrier increase as more layers are added. Increasing depth leads to higher barrier values until it saturates (as seen for VGG and ResNet).

and convolutional architectures. MLP architectures hit their peak at a lower width compared to CNNs and a decreasing trend starts earlier. For ResNets the barrier size is saturated at a high value and does not change. The barrier value for VGG architecture on different datasets is also saturated at a high value and does not change by increasing the width. Such similar behavior observed for both ResNets and VGG architectures is due to the effect of depth as discussed in the next paragraph.

**Depth:** We vary network depth in Figure 3 to evaluate its impact on the barrier between optimal solutions obtained from different initializations. For MLPs, we fix the layer width at $2^{10}$ while adding identical layers as shown along the x-axis. We observe a fast and significant barrier increase as more layers are added. For VGG architecture family we observe significant barriers. This might be due to the effect of convolution or depth. In order to shed light on this observation, we use Shallow CNN (Neyshabur, 2020) with only two convolutional layers. As can be seen in Figure 3 when Shallow CNN has two layers the barrier size is low, while keeping the layer width fixed at $2^{10}$ and adding more layers increases the barrier size. For residual networks we also consider three ResNet architectures with 18, 34 and 50 layers and observe the same barrier sizes as VGG for all these depth values. The main overall observation from depth experiments is that for both fully-connected and convolutional architectures, increasing depth increases the barrier size significantly so the effect of depth is not similar to width. This can also be attributed to the observations that deeper networks usually have a less smooth landscape (Li et al., 2017).

**Task difficulty and architecture choice:** In Figure 4 we look into the impact of the task difficulty provided by the dataset choice (MNIST, SVHN, CIFAR-10, CIFAR-100, and ImageNet (Deng et al., 2009)) and the architecture type (one-layer MLP with $2^{10}$ neurons, Shallow CNN with two convolutional layer and width of $2^{10}$, VGG-16 with batch-normalization, ResNet18 and ResNet50). Each row in Figure 4a and Figure 4b shows the effect of task difficulty, *e.g.*, fixing the task to SVHN and moving from MLP to Shallow CNN gives lower test error hence lower barrier size. Each column also represents the effect of architecture on a specific dataset, *e.g.*, fixing the architecture to Shallow CNN and moving from CIFAR10 to CIFAR100 presents an increase in test error, hence increase in the barrier size. Although deep architectures like VGG16 and ResNet18 present low test error, the discussed effect of depth saturates their barrier at a high level. Figure 4c aggregates the correlation between test error and size of the barrier. For MLP and Shallow CNN we observe a high positive correlation between test error and barrier size across different datasets. Deeper networks (VGGs, ResNets) form a cluster in the top-left, with low test error and high barrier size.

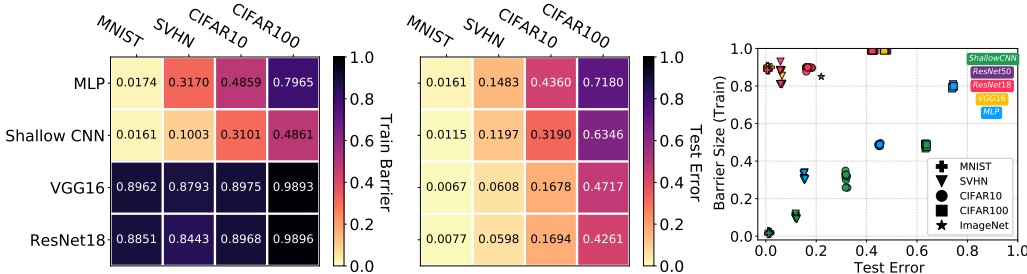

(a) Barrier for different architectures and task difficulty

(b) Achieved test error

(c) Lower test error results in lower barrier for shallow networks

Figure 4: **Effect of architecture choice and task difficulty on barrier size.** Each row in Figure 4a and Figure 4b shows the effect of task difficulty while each column represents the effect of architecture on a specific dataset. Figure 4c notes that a pair of (architecture, task) has lower barrier if the test error is lower. Therefore, any changes in the architecture or the task that improves the test error, also improves the loss barrier. Effect of depth is stronger than (architecture, task) which leads to high barrier values for ResNets on MNIST, SVHN, CIFAR10, CIFAR100, and ImageNet.

# 3 ROLE OF INVARIANCE IN LOSS BARRIERS

Understanding the loss landscape of deep networks has proven to be very challenging. One of the main challenges in studying the loss landscape without taking the optimization algorithm into account is that there exist many minima with different generalization properties. Most of such minima are not reachable by SGD and we only know about their existence through artificially-made optimization algorithms and training regimes (Neyshabur et al., 2017). To circumvent this issue, we focus on parts of the landscape that are reachable by SGD. Given a dataset and an architecture, one could define a probability distribution over all solutions reachable by SGD and focus on the subset where SGD is more likely to converge to.

## 3.1 INVARIANCES IN NEURAL NETWORK FUNCTION CLASS

We say that a network is *invariant* with respect to a transformation if and only if the network resulting from the transformation represents the same function as the original network. There are two well-known invariances: one is the unit-rescaling due to positive homogeneity of ReLU activations (Neyshabur et al., 2015) and the other is permutation of hidden units. Unit-rescaling has been well-studied and empirical evidence suggests that implicit bias of SGD would make the solution converge to a stage where the weights are more balanced (Neyshabur et al., 2015; Wu et al., 2019). Since we are interested in the loss landscape through the lens of SGD and SGD is much more likely to converge to a particular rescaling, consideration of this type of invariance does not seem useful. However, in the case of permutations, all permutations are equally likely for SGD and therefore, it is important to understand their role in the geometric properties of the landscape and its basins of attraction. Here we consider invariances that are in form of permutations of hidden units in each layer of the network, *i.e.*, each layer $i$ with parameters $W_i$ is replaced with $P_i W_i P_{i-1}$ where $P_i$ is a permutation matrix and $P_l = P_0$ is the identity matrix. Note that our results only hold for permutation matrices since only permutation commutes with nonlinearity. We use $\mathcal{P}$ to refer to the set of valid permutations for a neural network and use $\pi$ to refer to a valid permutation.

## 3.2 OUR CONJECTURE

As mentioned above, SGD's implicit regularization balances weight norms and, therefore, scale invariance does not seem to play an important role in understanding symmetries of solutions found by SGD. Consequently, here we focus on permutation invariance and conjecture that taking it into account allows us to have a much simpler view of SGD solutions. We first state our conjecture informally:

> *Most SGD solutions belong to a set $\mathcal{S}$ whose elements can be permuted in such a way that there is no barrier on the linear interpolation between any two permuted elements in $\mathcal{S}$.*

The above conjecture suggests that most SGD solutions end up in the same basin in the loss landscape after proper permutation (see Figure 1 left panel). We acknowledge that the above conjecture is bold.

Nonetheless, we argue that coming up with strong conjectures and attempting to disprove them is an effective method for scientific progress. Note, our conjecture also has great practical implications for model ensembling and parallelism since one can average models that are in the same basin in the loss landscape. The conjecture can be formalized as follows:

**Conjecture 1.** *Let $f(\theta)$ be the function representing a feedforward network with parameters $\theta \in \mathbb{R}^k$, $\mathcal{P}$ be the set of all valid permutations for the network, $P : \mathbb{R}^k \times \mathcal{P} \to \mathbb{R}^k$ be the function that applies a given permutation to parameters and returns the permuted version, and $B(\cdot, \cdot)$ be the function that returns barrier value between two solutions as defined in Equation 1. Then, there exists a width $h > 0$ such that for any network $f(\theta)$ of width at least $h$ the following holds: There exist a set of solutions $\mathcal{S} \subseteq \mathbb{R}^k$ and a function $Q : \mathcal{S} \to \mathcal{P}$ such for any $\theta_1, \theta_2 \in \mathcal{S}$, $B(P(\theta_1, Q(\theta_1)), \theta_2) \approx 0$ and with high probability over an SGD solution $\theta$, we have $\theta \in \mathcal{S}$.*

Next, we approach Conjecture 1 from both theoretical and empirical aspects and provide some evidence to support it.

### 3.3 A THEORETICAL RESULT

In this section we provide elementary theoretical results in support of our conjecture. Although the theoretical result is provided for a very limited setting, we believe it helps us understand the mechanism that could give rise to our conjecture. Bellow, we theoretically show that Conjecture 1 holds for a fully-connected network with a single hidden layer at initialization. Proof is given in Appendix D.

**Theorem 3.1.** *Let $f_{\mathbf{v}, \mathbf{U}}(\mathbf{x}) = \mathbf{v}^\top \sigma(\mathbf{U}\mathbf{x})$ be a fully-connected network with $h$ hidden units where $\sigma(\cdot)$ is ReLU activation, $\mathbf{v} \in \mathbb{R}^h$ and $\mathbf{U} \in \mathbb{R}^{h \times d}$ are the parameters and $\mathbf{x} \in \mathbb{R}^d$ is the input. If each element of $\mathbf{U}$ and $\mathbf{U}'$ is sampled uniformly from $[-1/\sqrt{d}, 1/\sqrt{d}]$ and each element of $\mathbf{v}$ and $\mathbf{v}'$ is sampled uniformly from $[-1/\sqrt{h}, 1/\sqrt{h}]$, then for any $\mathbf{x} \in \mathbb{R}^d$ such that $\|\mathbf{x}\|_2 = \sqrt{d}$, with probability $1 - \delta$ over $\mathbf{U}, \mathbf{U}', \mathbf{v}, \mathbf{v}'$, there exist a permutation such that*

$$\left| f_{\alpha\mathbf{v} + (1-\alpha)\mathbf{v}'', \alpha\mathbf{U} + (1-\alpha)\mathbf{U}''}(\mathbf{x}) - \alpha f_{\mathbf{v}, \mathbf{U}}(\mathbf{x}) - (1-\alpha) f_{\mathbf{v}', \mathbf{U}'}(\mathbf{x}) \right| = \tilde{O}(h^{-\frac{1}{2d+4}})$$

*where $\mathbf{v}''$ and $\mathbf{U}''$ are permuted versions of $\mathbf{v}'$ and $\mathbf{U}'$.*

Theorem 3.1 states that for wide enough fully-connected networks with a single hidden layer, one can find a permutation that leads to having no barrier at random initialization. Although, our prove only covers random initialization, we believe with a more involved proof, it might be possible to extend it to NTK regime (Jacot et al., 2018). We leave this for future work.

### 3.4 DIRECT EMPIRICAL EVALUATION OF CONJECTURE 1

Another possible approach is to use brute-force (BF) search mechanism and find the function $Q$ for elements of $\mathcal{S}$. The factorial growth of the number of permutations with the size of hidden units in each layer hinders exhaustive search for a winning permutation $\pi$ to linear mode connect $P(\theta_1, \pi)$ and $\theta_2$. Even for MLPs with just one hidden layer brute-force works in reasonable time up to $2^4$ neurons only, forcing the search to examine $2^4! \approx 2 \cdot 10^{13}$ permuted networks. BF is not feasible even for modest size deep networks. For small networks, one can use BF to find permutations between different models (see E.3). However, small size networks are not the focus of this paper and Conjecture 1 specifically mentions that.

Given the size of search space, using a more advanced search algorithm can be useful. The issue with this approach is that since it relies on the strength of a search algorithm, if the search algorithm fails in finding the permutation, one cannot be sure about the source of failure being the search algorithm or nonexistence of a permutation that leads to no barrier.

### 3.5 OUR MODEL VS REAL WORLD: AN ALTERNATIVE APPROACH

We propose the following approach to circumvent the above obstacles. We create a competing set $\mathcal{S}'$ (our model) as a proxy for set $\mathcal{S}$ (real world). Given an SGD solution $\theta_1 \in \mathcal{S}$, we define $\mathcal{S}' = \{P(\theta_1, \pi) | \forall \pi \in \mathcal{P}\}$. We know that set $\mathcal{S}'$ satisfies the conjecture. For set $\mathcal{S}'$, all points are

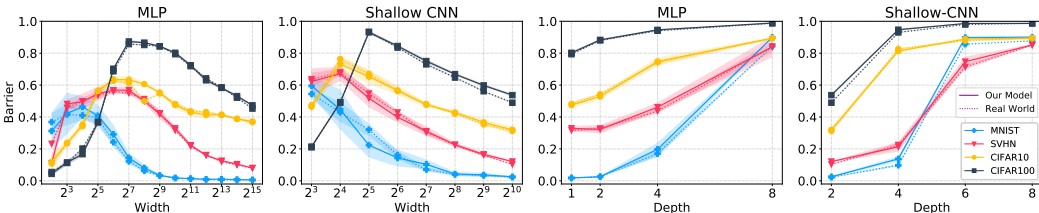

Figure 5: **Similar loss barrier between real world and our model *BEFORE* applying permutation.** From **left to right**: one-layer MLP, two-layer Shallow CNN, MLP and Shallow CNN with layer width of $2^{10}$. Increasing width first increases and then decreases the barrier, while adding more layers significantly increases the barrier size. We observe that $\mathcal{S}'$ and $\mathcal{S}$ behave similarly in terms of barrier as we change different architecture parameters such as width, depth across various datasets.

known permutations of $\theta_1$. Therefore, one can permute all points to remove their barriers with $\theta_1$, i.e, for all $\theta_2 = P(\theta_1, \pi)$, one can use $Q(\theta_2) = \pi^{-1}$ to remove the barrier between $\theta_1$ and $\theta_2$. Our goal is therefore to show that $\mathcal{S}$ is similar to $\mathcal{S}'$ in terms of barrier behavior.

Equivalence of $\mathcal{S}'$ and $\mathcal{S}$ in terms of barriers, means that if we choose an element $\theta$ in $\mathcal{S}$, one should be able to find permutations for each of other elements of the $\mathcal{S}$ so that the permuted elements have no barrier with $\theta$ and hence are in the same basin as $\theta$. The consequence of the equivalence of our model to real world is that Conjecture 1 holds. The conjecture effectively means that different basins exist because of the permutation invariance and if permutation invariance is taken into account (by permuting solutions to remove the barriers between them), there is only one basin, i.e., all solutions reside in the same basin in the loss landscape. We actually want to show $\mathcal{S}$ is similar to $\mathcal{S}'$ in terms of optimizing over all permutations but that is not possible so we show $\mathcal{S}$ is similar to $\mathcal{S}'$ in terms of barrier without search or when we search over a smaller set of permutations using a search algorithm. In the next section, we investigate our conjecture using this approach.

## 4 EMPIRICAL INVESTIGATION

In this section we show that $\mathcal{S}$ and $\mathcal{S}'$ have similar loss barrier along different factors such as width, depth, architecture, dataset and other model parameters (with and without searching for a permutation that reduces the barrier), hence supporting our conjecture. As discussed in Section 2, the barrier for both VGG and ResNet architectures is saturated at a high value hinting that the loss landscape might be more complex for these architecture families. In our experiments we observed that $\mathcal{S}$ and $\mathcal{S}'$ have similar high barriers for both of these architectures (see Appendix E.2). Moreover, we observed that for both $\mathcal{S}$ and $\mathcal{S}'$ the employed algorithms (Section 4.2) were unable to find a permutation to reduce the barrier and hence our model shows a similar behavior to real world [3]. Given that the width and depth do not influence the barrier behavior in VGG and ResNet architectures, here we only focus on the effect of width and depth on barrier sizes for MLPs and Shallow CNNs.

### 4.1 SIMILARITY OF $\mathcal{S}$ AND $\mathcal{S}'$

Figure 5 compares our model to the real world and shows that $\mathcal{S}'$ and $\mathcal{S}$ have strikingly similar barriers as we change different architecture parameters such as width and depth across various architecture families and datasets. This surprising level of similarity between our model and real world on variety of settings provide strong evidence for the conjecture. Even if the conjecture is not precisely correct as stated, the empirical results suggest that the structural similarities between our model and real world makes our model a useful simplification of the real world for studying the loss landscape. For example, the effect of width and depth on the barrier is almost identical in our model and the real world which suggests that permutations are perhaps playing the main role in such behaviors.

### 4.2 SEARCH ALGORITHMS FOR FINDING A WINNING PERMUTATION

The problem of finding a winning permutation $\pi \in \mathcal{P}$ is a variant of the Travelling Salesman Problem where neurons are mapped to cities visited by a salesman. The problem belongs to the class of NP-hard optimization problems and *simulated annealing* (SA) is often used to find a solution for such a combinatorial search problem. SA's performance however highly depends on the parameter

---

[3]All the experiments are included in the aggregated results reported in Figure 1.

---

**Algorithm 1** Simulated Annealing (SA) for Permutation Search

---

1: **procedure** SA($\{\theta_i\}, i = 1..n, n \geq 2$)         ▷ Goal: minimize the barrier between $n$ solutions
2:     $\pi_i = \pi_0, \forall i = 1..n$
3:     **for** $k = 0; k < k_{max}; k$++ **do**
4:         $T \leftarrow$ temperature($\frac{k+1}{k_{max}}$)
5:         Pick random candidate permutations $\{\hat{\pi}_i\}, \forall i = 1..n$
6:         **if** $\Psi(P(\theta_i, \hat{\pi}_i)) < \Psi(P(\theta_i, \pi_i))$ **then**         ▷ $\Psi$: barrier objective function
7:             $\pi_i \leftarrow \hat{\pi}_i$
        **return** $\{\pi_i\}$

---

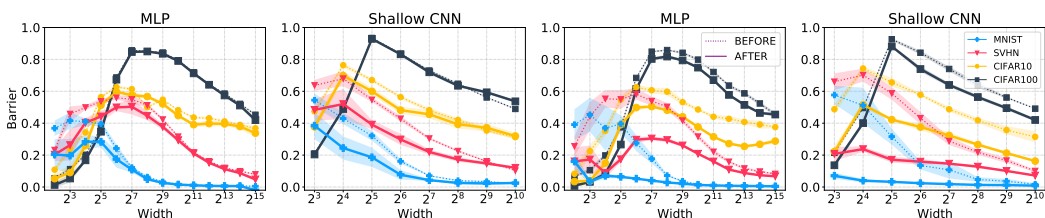

Figure 6: **Performance of Simulated Annealing (SA)**. **Two Left:** $SA_2$ where we average the weights of permuted models first and $\psi$ is defined as the train error of the resulting average model. **Two Right:** Search space is reduced *i.e.*, we take two SGD solutions $\theta_1$ and $\theta_2$, permute $\theta_1$ and report the barrier between permuted $\theta_1$ and $\theta_2$ as found by SA with $n = 2$. When search space is reduced, SA is able to find better permutations.

choices, including the minimum and maximum temperatures, the cooling schedule and the number of optimization steps. The pseudocode of SA is shown in Algorithm 1. SA takes a set of solutions $\{\theta_i\}, i = 1..n, n \geq 2$ as input (we use $n = 5$) and searches for a set of permutations $\{\pi_i\}$ that reduce the barriers between all permuted $\binom{n}{2}$ solution pairs. To find the best $\{\pi_i\}$, in each step of SA the current candidate permutations $\{\hat{\pi}_i\}, i = 1..n$ are evaluated to minimize the objective function $\Psi$. We use two versions of simulated annealing that vary in their definition of $\Psi$ to evaluate the conjecture.

**Simulated Annealing 1 ($SA_1$).** In the first version $SA_1$, $\Psi$ is defined as the average pairwise barrier between candidate permutations $B(P(\theta_i, \pi_i), P(\theta_j, \pi_j)), i \neq j$.

**Simulated Annealing 2 ($SA_2$).** In the second version $SA_2$, we average the weights of permuted models $P(\theta_i, \pi_i)$ first and defined $\Psi$ as the train error of the resulting average model. The simplest form of $SA_2$ happens if $n = 2$ and is discussed in Section A.3.

The rationale behind these two versions is that if the solutions reside in one basin, there is no barrier between them. Therefore averaging solutions in one basin yields another solution inside their convex hull. Although each version of SA has a different definition of the objective function $\Psi$ to find the best permutation, for all SA versions we report the average barrier between all pairs in the plots. Our empirical results suggest that $SA_1$ and $SA_2$ yield very similar performance. However, $SA_2$ is significantly less computationally expensive, which makes it more suitable for exploring larger models. In the following sections we present the results obtained with $SA_2$ only and refer to this version as SA. For more details on SA implementation see Appendix A.4. The left two plots in Figure 6 show that $SA_2$ is not able to find permutations that improve pair-wise barrier significantly. We know that SA does not guarantee finding a solution and is known to lose its effectiveness on TSP benchmarks beyond 1'000 cities (Zhan et al., 2016). The effectiveness of SA is also reduced here as we can only evaluate the cost of full route (divide and conquer is not possible). One way to increase this effectiveness is to reduce the search space which we will discuss next.

**Search space reduction**. In order to reduce the search space, here we only take two SGD solutions $\theta_1$ and $\theta_2$, permute $\theta_1$ and report the barrier between permuted $\theta_1$ and $\theta_2$ as found by SA with $n = 2$. The right two plots in Figure 6 shows this intervention helps SA to find better permutations. In particular, the barrier improves significantly for MNIST and SVHN datasets for both MLP and Shallow CNN across different width. However, similar to Section 2, we did not observe significant improvements when increasing depth (see Figure 12).

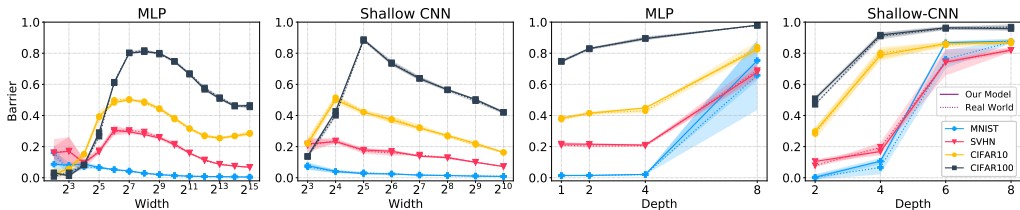

Figure 7: **Similar loss barrier between real world and our model *AFTER* applying permutation, when search space is reduced.** We observe that reducing the search space makes SA more successful in finding the permutation to remove the barriers. Specifically, SA could indeed find permutations that when applied to $\theta_1$ result in zero barrier *e.g.*, MLP for MNIST where depth is 1 (across all width), 2 and 4 (where width is $2^{10}$)

### 4.3 SIMILARITY OF $\mathcal{S}$ AND $\mathcal{S}'$ AFTER SEARCH

Figure 7 shows the surprising similarity of the barrier between $\mathcal{S}$ and $\mathcal{S}'$ even after applying a permutation found by a search algorithm (when search space is reduced). We also observe that reducing the search space makes SA more successful in finding the permutation $\{\pi\}$ to remove the barriers. Specifically, in some cases SA could indeed find permutations that when applied to $\theta_1$ result in zero barrier. However, the fact that SA's success shows a similar pattern for $\mathcal{S}, \mathcal{S}'$ provides another evidence in support of the conjecture. For example, SA successfully reduces the barrier for both $\mathcal{S}, \mathcal{S}'$ on MNIST and SVHN datasets. Figure 1 (right) summarizes our extensive empirical evidence (more than 3000 trained networks) in one density plot, supporting similarity of barriers in real world and our model across different choices of architecture family, dataset, width, depth, and random seed. Putting together, all our empirical results support our main conjecture.

## 5 DISCUSSIONS AND CONCLUSION

We investigated the loss landscape of ReLU networks, proposed and probed the conjecture that the barriers in the loss landscape between different solutions of a neural network optimization problem are an artifact of ignoring the permutation invariance of the function class. In a nutshell, this conjecture suggests that if one considers permutation invariance, there is essentially no loss barrier between different solutions and they all exist in the same basin in the loss landscape. Our analysis has direct implication on initialization schemes for neural networks. Essentially it postulates that randomness in terms of permutation does not impact the quality of the final result. It is interesting to explore whether it is possible to come up with an initialization that does not have permutation invariance and only acts like a perturbation to the same permutation. If all basins in the loss landscape are basically the same function, there will be no need to search all of them. One can explore the same basin while looking for diverse solutions and this makes search much easier and would lead to substantially more efficient search algorithms.

Another area where our analysis is of importance is for ensembles and distributed training. Related works (Frankle et al., 2020; Fort et al., 2019) show that simply averaging two SGD solutions would fail. If these models lie at the periphery of a wide and flat low loss region then ensembling them in their weight space (averaging), creates a model tending to the center of the region, which leads to performance improvement (Izmailov et al., 2019; Wen et al., 2020). If we can track the optimal permutation (that brings all solutions to one basin), it is possible to use it to do weight averaging and build ensembles more efficiently. Moreover, we are interested in answering the question whether there is a one-to-one mapping between lottery tickets and permutations. Frankle et al. (2020) requires stability to find lottery tickets. They define stability as the point in training trajectory where, if we branch at this point and train two copies with different seeds, the trained solutions are linearly mode connected. We conjecture that all the SGD trained solutions are linearly mode connected if the permutation is considered (satisfying the necessary condition for Lottery Ticket Hypothesis). We believe our analysis laid the ground for investigating these important questions and testing the usefulness of our conjecture in ensemble methods and pruning, which is the subject of future studies.

The biggest limiting factor of our study is the size of the search space and hence we need a strong search algorithm, specially for deep models where the size and complexity of the search space was prohibitive in terms of computation for the existing search methods. We hope improvements of search algorithms can help us to extend these results. Lastly, our analysis focuses on image recognition task and extending the results to natural language tasks is of interest for future work.

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

# APPENDIX

# A    IMPLEMENTATION DETAILS

We used Caliban (Ritchie et al., 2020) to manage all experiments in a reproducible environment in Google Cloud's AI Platform. Each point in plots show the mean value taken over 10 different runs (20 trained networks).

## A.1    TRAINING HYPER-PARAMETERS

Table 1 summarizes the set of used hyper-parameters for training different networks.

| Hyper-parameters | MLP | Shallow CNN | VGG | ResNet |
|---|---|---|---|---|
| Learning Rate | Fixed 0.01[1], Fixed 0.001[2] | Cosine 0.02 | Cosine 0.02 | Cosine 0.02 |
| Batch Size | 64 | 256 | 256 | 256 |
| Epochs | 3000 | 1000 | 1000 | 1000 |
| Momentum | 0.9 | 0.9 | 0.9 | 0.9 |
| Weight Decay | - | - | - | - |
| Data Augmentation | Normalization | Normalization | Normalization | Normalization |

[1] MNIST
[2] SVHN, CIFAR10, CIFAR100

Table 1: Training Hyper-parameters

## A.2    PERFORMANCE EVALUATION OF TRAINED MODELS: ERROR AND LOSS

Figure 8 demonstrate Train and Test error/loss for experiments on the effect of width. Here for MLP we have one hidden layer, for Shallow CNN two convolutional layer, ResNet is fixed to ResNet18, and VGG16 is selected from VGG family.    Figure 9 also demonstrates Train and Test error/loss for different depth. We set MLP to have 1024 hidden units in each layer, Shallow CNN, VGG and ResNest to have 1024, 64, 64 channels in each convolutional layer, respectively.

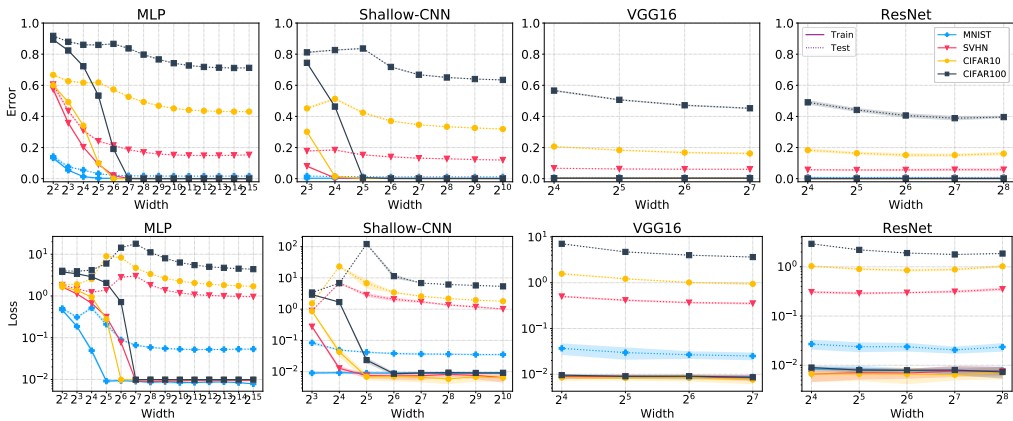

Figure 8: **Train and Test Error/Loss for Width**. Solid and dotted lines correspond to train and test respectively. All models are trained for 1000 epochs except MLP networks that are trained for 3000 epochs. The stopping criteria is either Cross Entropy Loss reaching 0.01 or number of epochs reaching maximum epochs.

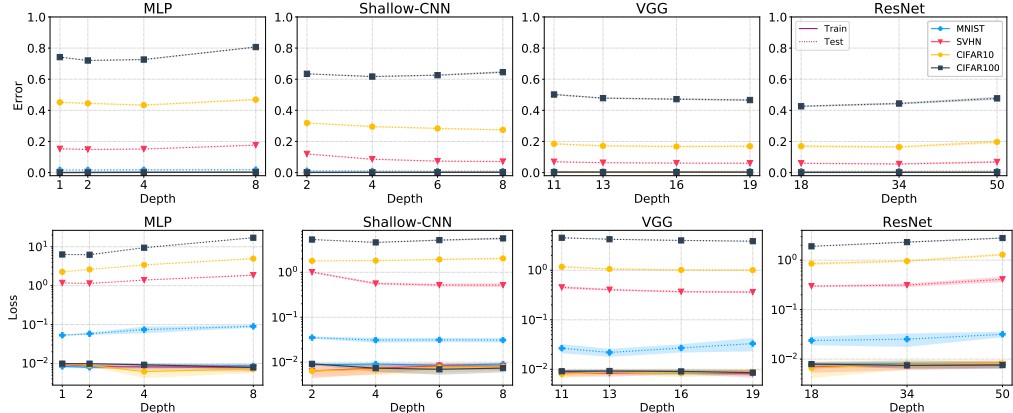

Figure 9: **Train and Test Error/Loss for Depth**. Solid and dotted lines correspond to train and test respectively. All models are trained for 1000 epochs except MLP networks that are trained for 3000 epochs. The stopping criteria is either Cross Entropy Loss reaching 0.01 or number of epochs reaching maximum epochs.

## A.3 SIMILARITY OF $\mathcal{S}$ AND $\mathcal{S}'$: DETAILED VIEW

**Aggregated empirical results**. Figure 10 shows the aggregation of our extensive empirical evidence (more than 3000 trained networks) in one plot comparing barriers in real world against our model across different choices of architecture family, dataset, width, depth, and random seed. Points with solid edges correspond to lowest barrier found after searching in the space of valid permutations using a Simulated Annealing (SA). SA shows better performance for shallow networks pushing more solid edge points to the lower left corner.

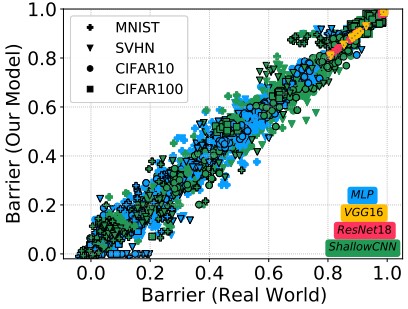

Figure 10: **Aggregation of empirical evidence on similarity of Real World and Our Model**. aggregation of our extensive empirical evidence (more than 3000 trained networks) in one plot comparing barriers in real world against our model across different choices of architecture family, dataset, width, depth, and random seed.

**Simulated Annealing performance**. In this Section, we show that $\mathcal{S}$ and $\mathcal{S}'$ have similar loss barriers we change different architecture parameters such as as width, depth across various datasets. Here we consider the mean over all pair-wise barriers as barrier size. Figure 5 shows the similarity of $\mathcal{S}$ and $\mathcal{S}'$ before permutation. In each plot, barrier size for $\mathcal{S}$ is similar to $\mathcal{S}'$. Such similarity is also observed in Figure 11, where we see the barrier after permutation using SA.

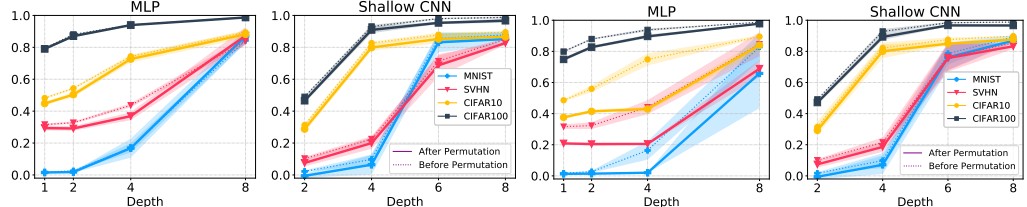

Figure 12: **Performance of Simulated Annealing (SA)**. **Left:** $SA_2$ where we average the weights of permuted models first and $\psi$ is defined as the train error of the resulting average model. **Right:** Search space is reduced *i.e.*, we take two SGD solutions $\theta_1$ and $\theta_2$, permute $\theta_1$ and report the barrier between permuted $\theta_1$ and $\theta_2$ as found by SA with $n = 2$. When search space is reduced, SA is able to find better permutations.

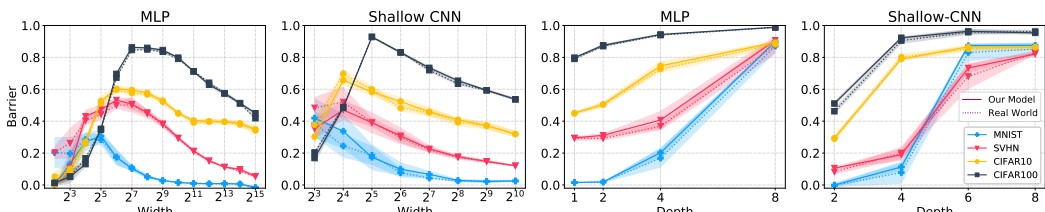

Figure 11: **Similar loss barrier between real world and our model** *after* **applying permutation.** Effects of width and depth also holds in this setting. Compared to Figure 5, we observe slight barrier reduction. Reducing search space helps SA to find better solutions (see section A.3).

**Search space reduction**. In order to reduce the search space, here we only take two SGD solutions $\theta_1$ and $\theta_2$, permute $\theta_1$ and report the barrier between permuted $\theta_1$ and $\theta_2$ as found by SA with $n = 2$. Figure 13 and Figure 7 show the effect of width and depth on barrier similarity between $\mathcal{S}$ and $\mathcal{S}'$ before and after permutation. Comparing Figure 13 and Figure 7 shows that SA succeeds in barrier removal. SA performance on $\mathcal{S}$ and $\mathcal{S}'$ yields similar results for both before and after permutation scenarios. Such similar performance is observed along a wide range of width and depth for both MLP and Shallow-CNN over different datasets (MNIST, SVHN, CIFAR10, CIFAR100). We look into effects of changing model size in terms of width and depth as in earlier sections, and note that similar trends hold for before and after permuting solution $\theta_1$. Comparing Figure 13 and Figure 7 shows that reducing the search space makes SA more successful in finding the permutation $\{\pi\}$ to remove the barriers. Specifically, SA can indeed find permutations across different networks and datasets that result in zero barrier when applied to $\theta_1$. SA can also find permutations that reduce the barrier for both MLP and Shallow-CNN across different width, depth and datasets. For example, such cases include MLP for MNIST, SVHN, CIFAR10, and CIFAR100 where depth is 1 and width is $2^3$ and $2^4$, MLP for MNIST where depth is 2 and width is $2^{10}$, Shallow-CNN for MNIST, SVHN, CIFAR10, CIFAR100 where depth is 2 and width is $2^4$ and for Shallow-CNN for MNIST where depth is 2 and width is $2^6$.

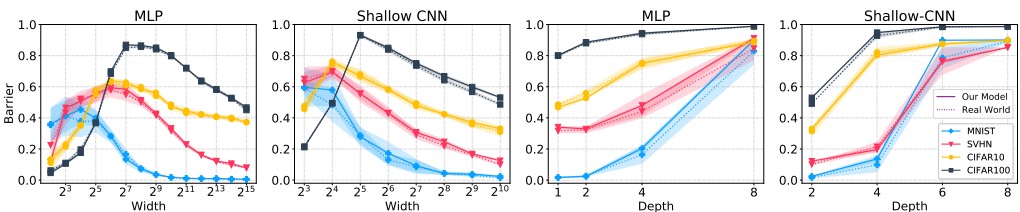

Figure 13: **Effect of width and depth on barrier similarity between real world and our model** *before* **permutation**. Search space is reduced here *i.e.*, we take two SGD solutions $\theta_1$ and $\theta_2$, permute $\theta_1$ and report the barrier between permuted $\theta_1$ and $\theta_2$ as found by SA with $n = 2$. Similarity of loss barrier between real world and our model is preserved across model type and dataset choices as width and depth of the models are increased.

### A.4   SIMULATED ANNEALING

We use Simanneal[4] as python module for simulated annealing. The process involves:

- Randomly move or alter the state (generate a permutation)

- Assess the energy of the new state (permuted model) using the objective function (Linear Mode Connectivity based on Equation 1)

- Compare the energy to the previous state and decide whether to accept the new solution or reject it based on the current temperature.

For a move to be accepted, it must meet one of two requirements:

- The move causes a decrease in state energy (i.e. an improvement in the objective function)

- The move increases the state energy (i.e. a slightly worse solution) but is within the bounds of the temperature.

**Temperature**. In each step, the generated permutation is chosen with the probability of $P = e^{\frac{-cost}{temperature}}$, where cost is the barrier at $\alpha = \frac{1}{2}$. In the first steps, as the temperature is high, there is a high probability that the worse neighbor is also selected. The neighbor is another permutation that if applied, differs slightly in the order of the neurons/channels. As we move forward the temperature decreases with $T = e^{e^{\frac{-Tmax \times steps}{Tmin \times steps}}}$ and we stick to permutations that improve the barrier.

**Scaling the computation for SA**. In an experiment, we scale number of steps in simulated annealing to investigate the effect of this hyper-parameter. If the barrier continues to decrease as the amount of computation increases and does not plateau, then this would suggest that with enough computation, the barrier found by simulated annealing could eventually go to zero. Figure 14 shows that increasing number of steps exponentially, helps SA to find better solutions. Table 3 shows that as the number of steps increases ($10\times$), $\Delta$ moves towards 2 *i.e.*, 50% reduction in barrier ($\Delta > 0$ means barrier does not plateau) . Running SA for 50K steps takes 10K seconds on an n1-standard-8 GCP machine (8 vCPU, 30 GB RAM) with 1xV100 GPU. Due to limited computational resources, we set number of the steps to 50K.

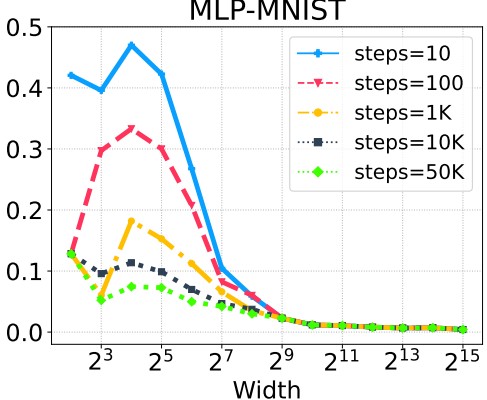

Figure 14: **Scaling cost for Simulated Annealing**. Increasing number of steps exponentially, helps SA to find better solutions. As the amount of computation increases the barrier continues to decrease.

---

[4]https://github.com/perrygeo/simanneal

| width=16 | | | width=32 | | | width=64 | | |
|---|---|---|---|---|---|---|---|---|
| steps | barrier | $\Delta$ barrier | steps | barrier | $\Delta$ barrier | steps | barrier | $\Delta$ barrier |
| 10 | 0.470 | - | 10 | 0.430 | - | 10 | 0.271 | - |
| 100 | 0.331 | $1.42\times$[1] | 100 | 0.302 | $1.43\times$ | 100 | 0.202 | $1.35\times$ |
| 1K | 0.190 | $1.73\times$ | 1K | 0.175 | $1.71\times$ | 1K | 0.121 | $1.41\times$ |
| 10K | 0.105 | $1.80\times$ | 10K | 0.995 | $1.75\times$ | 10K | 0.085 | $1.71\times$ |
| 50K | 0.055 | $1.90\times$ | 50K | 0.055 | $1.80\times$ | 50K | 0.048 | $1.77\times$ |

[1] $\Delta = \frac{0.470}{0.331} = 1.42$

Table 2: **Scaling cost for Simulated Annealing** As the amount of computation increases the barrier continues to decrease. As the number of steps increases, $\Delta$ increases toward 2 and does not plateau.

The following code runs simulated annealing to find the best permutation in Section 4.1

```
// Simulated Annealing
from simanneal import Annealer
def barrier_SA(arch, model, sd1, sd2, w2, init_state, tmax, tmin, steps, train_inputs,
    train_targets, train_avg_org_models, nchannels, nclasses, nunits):
    class BarrierCalculationProblem(Annealer):
        """annealer with a travelling salesman problem."""
        def __init__(self, state):
            super(BarrierCalculationProblem, self).__init__(state) # important!

        def move(self):
            """Swaps two cities in the route."""
            initial_energy = self.energy()
            for j in range(5):
                for i in range(len(self.state[j])):
                    x = self.state[j][i]
                    a = random.randint(0, len(x) - 1)
                    b = random.randint(0, len(x) - 1)
                    self.state[j][i][a], self.state[j][i][b] = self.state[j][i][b],
                        self.state[j][i][a]
            return self.energy() - initial_energy

        def energy(self):
            """Calculates the cost for proposed permutation."""
            permuted_models = []
            for i in range(5):
                permuted_models.append(permute(arch, model, self.state[i], sd2[i], w2[i],
                    nchannels, nclasses, nunits))
            #### form one model which is the average of 5 permuted models
            permuted_avg = copy.deepcopy(model)
            new_params = OrderedDict()
            for key in sd2[0].keys():
                param = 0
                for i in range(len(permuted_models)):
                    param = param + permuted_models[i][key]
                new_params[key] = param / len(permuted_models)
            permuted_avg.load_state_dict(new_params)
            eval_train = evaluate_model(permuted_avg, train_inputs, train_targets)['top1']
            cost = 1 - eval_train
            return cost

    bcp = BarrierCalculationProblem(init_state)
    bcp_auto = {'tmax': tmax, 'tmin': tmin, 'steps': steps, 'updates': 100}
    bcp.set_schedule(bcp_auto)
    winning_state, e = bcp.anneal()
    return winning_state
```

## B   IMPROVE SEARCH ALGORITHM

In the main text we use simulated annealing (SA) to find the winning permutation $\pi$. Figure 6 shows that SA is only able to reduce the barrier, and reducing search space (n = 2) helps in finding better permutations. However, a critical question still remains: Is there an algorithm that finds better permutations?

He et al. (2018) proposed an algorithms that merges correlated, pre-trained deep neural networks for cross-model compression. Their objective is to zip two neural networks, optimized for two different tasks, into one network. The ultimate network does both tasks without losing too much accuracy on each task. Their algorithm is based on layer-wise neuron sharing, which uses f(weights, post activations) to find which neurons could be zipped together. They define the similarity (or equivalently difference) of two neurons as below (Eq. 12 in the original paper):

$$\delta_{n_A, n_B} = \frac{1}{2}(w_{l,i}^A - w_{l,i}^B).((H_{l,i}^A)^{-1} + (H_{l,i}^B)^{-1})^{-1}.(w_{l,i}^A - w_{l,i}^B) \tag{2}$$

We also used their "functional difference" as a measure for neuron matching between two randomly initialized trained networks. They used post activation as an approximation for Hessian matrices. Calculating the difference between each pair of neurons based on Equation 2, gives an $m \times m$ matrix (m is width of the network). In the second step, neurons with minimum distance are matched together in a greedy way *i.e.*, if $n_{i,A}$ and $n_{j,B}$ have the minimum distance, row $i$ and column $j$ is removed from the distance matrix. Figure 15 shows that using functional difference, the barrier could be improved.

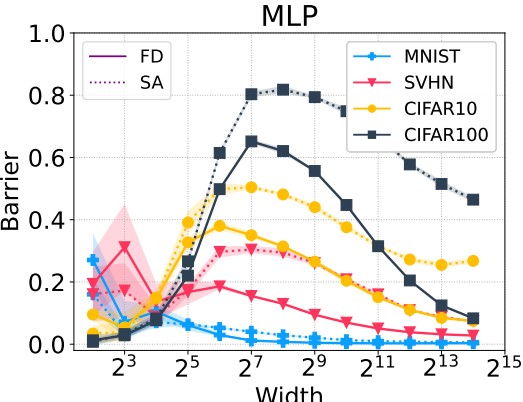

Figure 15: **Performance of Functional Difference compared to Simulated Annealing**. Functional Difference could indeed find better permutations, improving the barrier size between two solutions.

## C   MAKING ENSEMBLES

As stated before, our conjecture has implications for ensemble methods. If two solutions lie at the periphery of a wide and flat low loss region (a basin where models are linearly connected to each other), then ensembling them in their weight space (averaging), creates a model tending to the center of the region, which leads to performance improvement. However, Simulated Annealing could not find the optimal permutations for all cases. Section B shows that using matching algorithms like functional difference could help to find better permutations. Motivated to create ensembles, Wortsman et al. (2021) start with two (or more) random initializations and learn a subspace (a line or simplex) connecting them. Throughout training they sample one (or more) points on this line and add the loss at this point to the loss of training. In order to enforce diversity in function space, they also add a regularization term as cosine similarity of two endpoints of the line. However as they enforce two models to be in a line, the functional diversity of the final ensemble is limited compare to our methods. Here we combine their method with functional difference to make best of both worlds.

In this experiment, we train two randomly initialized networks separately, and then use functional difference to decrease the barrier between final solutions. Then we use learning subspace method to make them in one basin. Our results on MLP with one hidden layer and width of 1024 neurons on MNIST and CIFAR10 shows that this method outperforms the others.

| Architecture | Width | Dataset | FD[1] | SL[2] | FD + SL |
|---|---|---|---|---|---|
| MLP | 1024 | MNIST | 96.85 | 97.63 | **98.22** |
| MLP | 1024 | CIFAR10 | 52.95 | 57.89 | **58.94** |

[1] Functional Difference (He et al., 2018)
[2] Subspace Learning (Wortsman et al., 2021)

Table 3: **Performance comparison of ensemble methods.** While functional difference (He et al., 2018) gives better permutations compared to simulated annealing, Subspace Learning (Wortsman et al., 2021) enforces two solutions into one basin from scratch. We combine the best of two worlds in FD + SL to guarantee functional diversity of learned solutions and also make them in one basin.

## D    PROOF OF THEOREM 3.1

We first recap Theorem 3.1 below for convenience and then provide the proof

**Theorem D.1** (3.1). *Let $h$ be the number of hidden units, $d$ be the input size. Let the function $f_{\mathbf{v},\mathbf{U}}(\mathbf{x}) = \mathbf{v}^\top \sigma(\mathbf{U}\mathbf{x})$ where $\sigma(\cdot)$ is ReLU activation, $\mathbf{v} \in \mathbb{R}^h$ and $\mathbf{U} \in \mathbb{R}^{h \times d}$ are parameters and $\mathbf{x} \in \mathbb{R}^d$ is the input. We show that if each element of $\mathbf{U}$ and $\mathbf{U}'$ is sampled uniformly from $[-1/\sqrt{d}, 1/\sqrt{d}]$ and each element of $\mathbf{v}$ and $\mathbf{v}'$ is sampled uniformly from $[-1/\sqrt{h}, 1/\sqrt{h}]$, then for any $\mathbf{x} \in \mathbb{R}^d$ such that $\|\mathbf{x}\|_2 = \sqrt{d}$, with probability $1 - \delta$ over $\mathbf{U}, \mathbf{U}', \mathbf{v}, \mathbf{v}'$, there exist a permutation such that*

$$\left| f_{\alpha\mathbf{v}+(1-\alpha)\mathbf{v}'', \alpha\mathbf{U}+(1-\alpha)\mathbf{U}''}(\mathbf{x}) - \alpha f_{\mathbf{v},\mathbf{U}}(\mathbf{x}) - (1-\alpha) f_{\mathbf{v}',\mathbf{U}'}(\mathbf{x}) \right| = \tilde{O}(h^{-\frac{1}{2d+4}})$$

*where $\mathbf{v}''$ and $\mathbf{U}''$ are permuted versions of $\mathbf{v}'$ and $\mathbf{U}'$.*

*Proof.* For any given $\xi > 0$, we consider the set $S_\xi = \{-1/\sqrt{d} + \xi, -1/\sqrt{d} + 3\xi, \ldots, 1/\sqrt{d} - \xi\}^d$ which has size $(\frac{1}{\xi\sqrt{d}})^d$ [5]. For any $s \in S_\xi$, let $C_s(\mathbf{U})$ be the set of indices of rows of $\mathbf{U}$ that are closest in Euclidean distance to $s$ than any other element in $S_\xi$:

$$C_s(\mathbf{U}) = \{i | s = \arg\min_{s' \in S_\xi} \|\mathbf{u}_i - s'\|_\infty \} \tag{3}$$

where for simplicity we assume that $\arg\min$ returns a single element. We next use the function $C_s$ to specify a permutation that allows each row in $\mathbf{U}'$ to be close to its corresponding row in $\mathbf{U}$. For every $s \in S_\xi$, we consider a random matching of elements in $C_s(\mathbf{U})$ and $C_s(\mathbf{U}')$ and when the sizes don't match, add the extra items in $\mathbf{U}$ and $\mathbf{U}'$ to the sets $I$ and $I'$ accordingly to deal with them later.

Since each element of $\mathbf{U}$ and $\mathbf{U}'$ is sampled uniformly from $[-1/\sqrt{d}, 1/\sqrt{d}]$, for each row in $\mathbf{U}$ and $\mathbf{U}'$, the probability of being assigned to each $s \in S_\xi$ is a multinomial distribution with equal probability for each $s$. Given any $s \in S_\xi$, we can use Hoeffding's inequality to bound the size of $|C_s(\mathbf{U})|$ with high probability. For any $t \geq 0$:

$$P\left(\left||C_s(\mathbf{U})| - (h/|S_\xi|)\right| \geq t\right) \leq -2\exp(-2t^2/h) \tag{4}$$

By union bound over all rows of $\mathbf{U}$ and $\mathbf{U}'$, with probability $1 - \delta/3$, we have that for every $s \in S_\xi$,

$$\frac{h}{|S_\xi|} - \sqrt{\frac{h}{2}\log(12|S_\xi|/\delta)} \leq |C_s(\mathbf{U})|, |C_s(\mathbf{U}')| \leq \frac{h}{|S_\xi|} + \sqrt{\frac{h}{2}\log(12|S_\xi|/\delta)} \tag{5}$$

---

[5]For simplicity, we assume that $\xi$ is chosen so that $1/\sqrt{d}$ is a multiple of $\xi$.

Consider $I$ and $I'$ which are the sets of indices that we throw out during the index assignment because of the size mismatch. Then, based on above inequality, we have that with probability $1 - \delta/3$,

$$|I| = |I'| = \frac{1}{2} \sum_{s \in S_\xi} \big||C_s(\mathbf{U})| - |C_s(\mathbf{U}')|\big| \le |S_\xi|\sqrt{\frac{h}{2} \log(12|S_\xi|/\delta)} \tag{6}$$

We next randomly match the indices in $I$ and $I$. Let $\mathbf{U}''$ be the matrix after applying the permutation to $\mathbf{U}'$ that corresponds to above matching of rows of $\mathbf{U}'$ to their corresponding row in $\mathbf{U}$. Note that for any $i \in [h] \setminus I$, we have that $\|\mathbf{u}_i - \mathbf{u}_i''\|_\infty \le 2\xi$ and for $i \in I$, we have $\|\mathbf{u}_i - \mathbf{u}_i''\|_\infty \le 2/\sqrt{d}$. We next upper bound the left hand side of the inequality in the theorem statement:

$$\Big|f_{\alpha\mathbf{v}+(1-\alpha)\mathbf{v}'',\alpha\mathbf{U}+(1-\alpha)\mathbf{U}''}(\mathbf{x}) - \alpha f_{\mathbf{v},\mathbf{U}}(\mathbf{x}) - (1-\alpha)f_{\mathbf{v}',\mathbf{U}'}(\mathbf{x})\Big|$$

$$= \Big|(\alpha\mathbf{v} + (1-\alpha)\mathbf{v}'')^\top \sigma((\alpha\mathbf{U}(1-\alpha)\mathbf{U}'')\mathbf{x}) - \alpha\mathbf{v}^\top \sigma(\mathbf{U}\mathbf{x}) - (1-\alpha)\mathbf{v}''^\top \sigma(\mathbf{U}''\mathbf{x})\Big|$$

$$= \Big|\alpha\mathbf{v}^\top[\sigma((\alpha\mathbf{U} + (1-\alpha)\mathbf{U}'')\mathbf{x}) - \sigma(\mathbf{U}\mathbf{x})] + (1-\alpha)\mathbf{v}''^\top[\sigma((\alpha\mathbf{U} + (1-\alpha)\mathbf{U}'')\mathbf{x}) - \sigma(\mathbf{U}''\mathbf{x})]\Big|$$

$$\le \Big|\alpha\mathbf{v}^\top[\sigma((\alpha\mathbf{U} + (1-\alpha)\mathbf{U}'')\mathbf{x}) - \sigma(\mathbf{U}\mathbf{x})]\Big| \tag{7}$$

$$+ \Big|(1-\alpha)\mathbf{v}''^\top[\sigma((\alpha\mathbf{U} + (1-\alpha)\mathbf{U}'')\mathbf{x}) - \sigma(\mathbf{U}''\mathbf{x})]\Big|$$

Since each element of $\mathbf{v}$ is sampled uniformly from $[-1/\sqrt{h}, 1/\sqrt{h}]$, for any $\mathbf{r} \in \mathbb{R}^h$ we have that $\mathbb{E}[\mathbf{v}^\top \mathbf{r}] = 0$ and by Hoeffding's inequality,

$$P\left(\Big|\mathbf{v}^\top \mathbf{r}\Big| \ge t\right) \le 2\exp\left(\frac{-ht^2}{2\|\mathbf{r}\|_2^2}\right) \tag{8}$$

Using the above argument, with probability $1-\delta/3$, we can bound the right hand side of inequality (7) as follows:

$$\Big|\alpha\mathbf{v}^\top[\sigma((\alpha\mathbf{U} + (1-\alpha)\mathbf{U}'')\mathbf{x}) - \sigma(\mathbf{U}\mathbf{x})]\Big|$$

$$+ \Big|(1-\alpha)\mathbf{v}''^\top[\sigma((\alpha\mathbf{U} + (1-\alpha)\mathbf{U}'')\mathbf{x}) - \sigma(\mathbf{U}''\mathbf{x})]\Big|$$

$$\le \alpha\sqrt{\frac{2\log(12/\delta)}{h}}\big\|\sigma((\alpha\mathbf{U} + (1-\alpha)\mathbf{U}'')\mathbf{x}) - \sigma(\mathbf{U}\mathbf{x})\big\|_2 \tag{9}$$

$$+ (1-\alpha)\sqrt{\frac{2\log(12/\delta)}{h}}\big\|\sigma((\alpha\mathbf{U} + (1-\alpha)\mathbf{U}'')\mathbf{x}) - \sigma(\mathbf{U}''\mathbf{x})\big\|_2$$

$$\le \alpha\sqrt{\frac{2\log(12/\delta)}{h}}\big\|(\alpha\mathbf{U} + (1-\alpha)\mathbf{U}'')\mathbf{x} - \mathbf{U}\mathbf{x}\big\|_2 \tag{10}$$

$$+ (1-\alpha)\sqrt{\frac{2\log(12/\delta)}{h}}\big\|(\alpha\mathbf{U} + (1-\alpha)\mathbf{U}'')\mathbf{x} - \mathbf{U}''\mathbf{x}\big\|_2$$

$$= \alpha\sqrt{\frac{2\log(12/\delta)}{h}}\big\|(1-\alpha)(\mathbf{U} - \mathbf{U}'')\mathbf{x}\big\|_2 + (1-\alpha)\sqrt{\frac{2\log(12/\delta)}{h}}\big\|\alpha(\mathbf{U} - \mathbf{U}'')\mathbf{x}\big\|_2$$

$$\le \sqrt{\frac{\log(12/\delta)}{2h}}\big\|(\mathbf{U} - \mathbf{U}'')\mathbf{x}\big\|_2 \tag{11}$$

where the inequality 10 is due to Lipschitz property of ReLU activations. Now, all we need to do is to bound $\big\|(\mathbf{U} - \mathbf{U}'')\mathbf{x}\big\|_2$. Note that for any $(i, j) \in [h] \times [d]$, $u_{ij} - u_{ij}''$ is an independent random variable with mean zero and bounded magnitude($2/\sqrt{d}$ if $i \in I$ and $2\xi$ otherwise). Therefore, we can again use the Hoffding's inequality similar to inequality (8) for each row $i$ and after taking a union

bound, we have the following inequality with probability $1 - \delta/3$,

$$
\begin{aligned}
\left\| (\mathbf{U} - \mathbf{U}'')\mathbf{x} \right\|_2 &= \sqrt{ \sum_{i \in I} (\mathbf{u}_i - \mathbf{u}_i'')\mathbf{x} + \sum_{i \in [h] \setminus I} (\mathbf{u}_i - \mathbf{u}_i'')\mathbf{x} } \\
&\leq \|x\|_2 \sqrt{ |I| \frac{4 \log(12h/\delta)}{d} + (h - |I|)(4\xi^2 \log(12h/\delta)) } \\
&\leq 2\sqrt{ \log(12h/\delta) \left( |I| + \xi^2 dh \right) }
\end{aligned}
$$

Where the last inequality is using $\|\mathbf{x}\|_2 = \sqrt{d}$. Substituting the above inequality into the right hand side of the inequality (11), gives us the following upper bound on the left hand side of the inequality in the theorem statement:

$$
\begin{aligned}
&\left| f_{\alpha \mathbf{v} + (1-\alpha)\mathbf{v}'', \alpha \mathbf{U} + (1-\alpha)\mathbf{U}''}(\mathbf{x}) - \alpha f_{\mathbf{v}, \mathbf{U}}(\mathbf{x}) - (1 - \alpha)f_{\mathbf{v}', \mathbf{U}'}(\mathbf{x}) \right| \\
&\leq \sqrt{ \frac{\log(12/\delta)}{2h} } \left\| (\mathbf{U} - \mathbf{U}'')\mathbf{x} \right\|_2 \tag{12} \\
&\leq \sqrt{ 2 \log(12/\delta) \log(12h/\delta) \left( \frac{|I|}{h} + \xi^2 d \right) } \tag{13}
\end{aligned}
$$

Setting $\xi = \epsilon / \sqrt{4d \log(12/\delta) \log(12h/\delta)}$, gives the following bound on $h$:

$$
\begin{aligned}
h &\leq \frac{4 \log(12/\delta) \log(12h/\delta)|I|}{\epsilon^2} \\
&\leq \frac{4 \log(12/\delta) \log(12h/\delta)|S_\xi| \sqrt{\frac{h}{2} \log(12|S_\xi|/\delta)}}{\epsilon^2}
\end{aligned}
$$

Therefore, we have:

$$
\begin{aligned}
h &\leq \left( \frac{4 \log(12/\delta) \log(12h/\delta)|S_\xi| \sqrt{\log(12|S_\xi|/\delta)}}{\epsilon^2} \right)^2 \\
&\leq \left( \frac{4 \log(12/\delta) \log(12h/\delta)}{\epsilon^2} \right)^{d+2} (\log(12/\delta) + d \log(1/\epsilon))
\end{aligned}
$$

Using the above inequality, we have $\epsilon = \tilde{O}(h^{-\frac{1}{2d+4}})$. $\qquad \square$

## E ADDITIONAL PLOTS

### E.1 BARRIER BEHAVIOR UNDER NOISY LABELS

Related works (Zhang et al., 2017) show that neural nets can memorize random labels. In this section we want to see whether the barrier changes if one starts to inject random labels into the training dataset. Our results over 5 different runs show that the barrier size behavior does not change, however including higher level of noise in labels lead to small increase in barrier size.

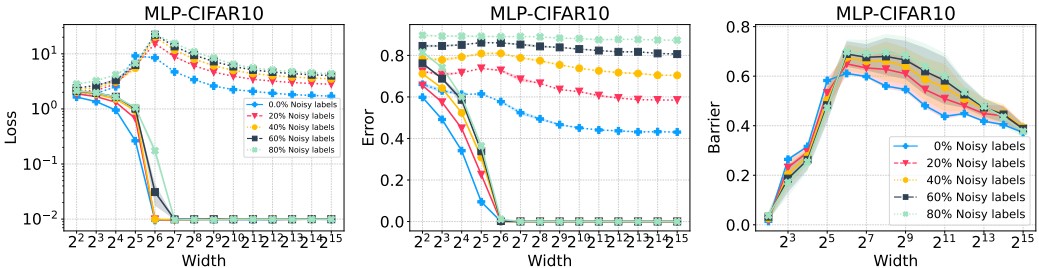

Figure 16: **Effect of label noise on barrier size. Left:** Train and Test Loss under different label noise. **Middle:** Train and Test Error under different label noise. **Right:** Train barrier (accuracy) under different label noise. Our results over 5 different runs show that the barrier size behavior does not change, however including higher level of noise in labels lead to small increase in barrier size.

## E.2 BARRIER: VGG AND RESNET

Figure 17 shows the barrier similarity for VGG and ResNet families between real world and our model. The left two panels shows the effect of width, while the right two panels illustrate the depth. As discussed in section 2, the barrier for VGGs and ResNets, for both real world and our model, is saturated at a high value and does not change.

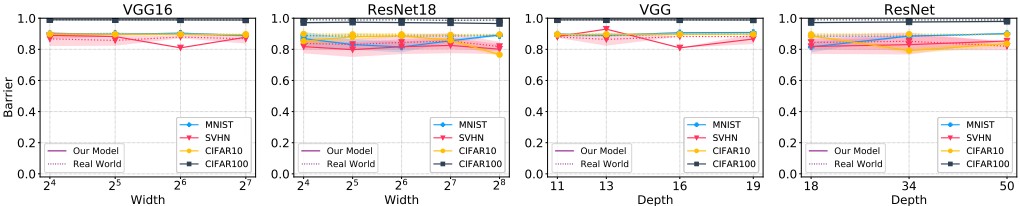

Figure 17: **Effect of width and depth on barrier similarity between real world and our model** *before* **permutation: VGG and ResNet families.** The left two panels shows the effect of width, while the right two panels illustrate the depth. As discussed in section 2, the barrier for VGGs and ResNets, for both real world and our model, is saturated at a high value and does not change.

## E.3 SMALL NETWORKS

We consider MLPs with the same architecture starting from 100 different initializations and trained on the MNIST dataset. For each pair of the networks we calculate their loss barrier and plot the histogram on the values. Next for each pair of the networks, we find the permutation that minimizes the barrier between them and plot the histogram for all the pairs. We do this investigation for different network sizes. The results are shown in Figure 18 top row. Note that since we consider all possible pairs, the observed barrier values are not i.i.d. If instead we randomly divide the 100 trained networks into two sets and choose pairs that are made by picking one network from each set, we will have an i.i.d sampling strategy. We investigate the pairs of networks trained on MNIST and measure the value of the direct and indirect barriers between them. *Indirect barrier* between two networks A, C is minimum over all possible intermediate points B of maximum of barrier between A, B and barrier between B, C, *i.e.*,

$$\min_{B != A, B != C} \max(B(\theta_A, \theta_B), B(\theta_B, \theta_C)).$$

The reason we look into this value is that if maximum between two barriers is small, it means that both barriers are small and therefore there exist an indirect path between A and C.

## E.4 LOSS BARRIERS ON THE TEST SET

**Width**. We evaluate the impact of width on the test barrier size in Figure 19. In comparison to Figure 2 the magnitude of test barriers are shifted to lower values as the test accuracy is lower than

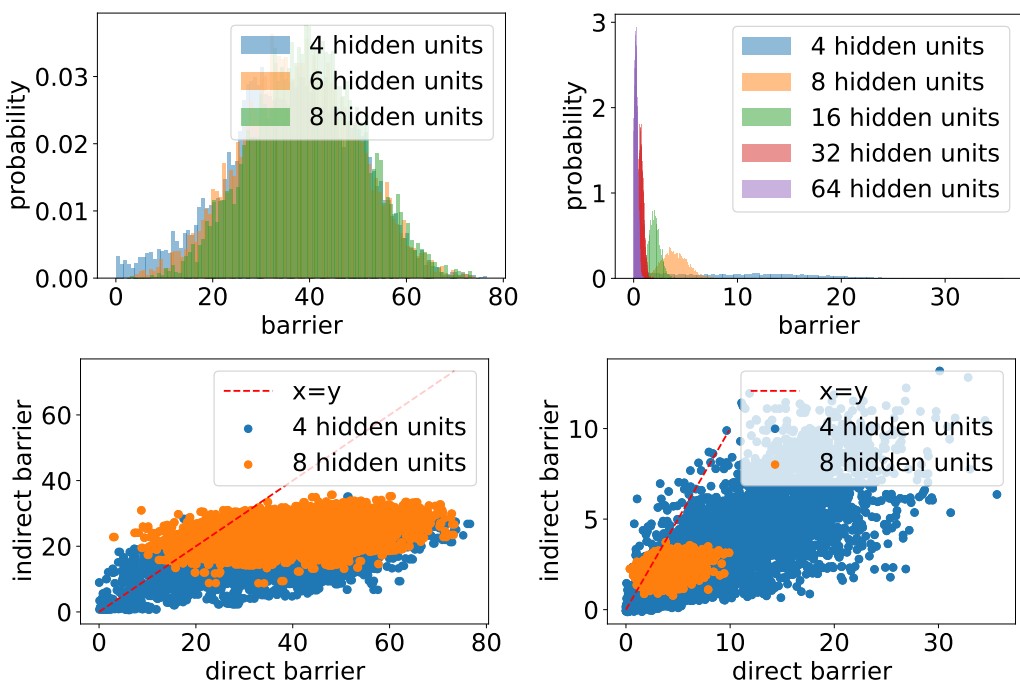

Figure 18: **Histogram of barrier values between pairs of 100 networks with the same architecture trained on MNIST starting from different initializations.** We find a permutation for each pair that minimizes the barrier. This is done for two layer MLPs and repeated for networks of different sizes. Bottom row: Indirect barrier between two networks A,C is minmimum over all possible intermediate points B of maximum of barrier between A, B and barrier between B, C. **Left:** before the permutation; **Right:** after the permutation.

train accuracy. This effect is intensified for harder tasks such as CIFAR100. The double descent phenomena is also observed here, especially for simpler tasks, *e.g.*, MNIST and SVHN.

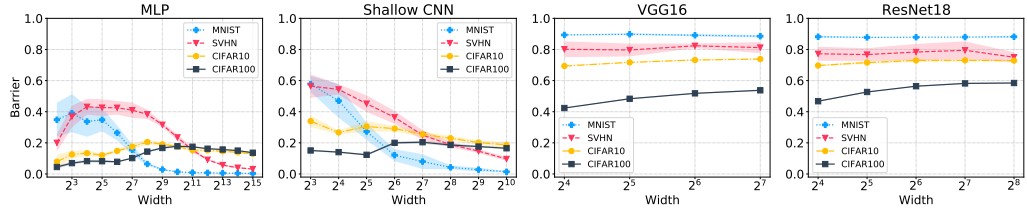

Figure 19: **Effect of width on barrier size (Test).** From **left to right**: one-layer MLP, two-layer Shallow-CNN, VGG-16, and ResNet-18 architectures and MNIST, CIFAR-10, SVHN, CIFAR-100 datasets. When the task is hard (CIFAR10, CIFAR100) the test barrier shrinks. For simpler tasks and large width sizes also the barrier becomes small.

**Depth**. We evaluate the impact of depth on the test barrier size in Figure 20. For MLPs, we fixed the layer width at $2^{10}$ while adding identical layers as shown along the x-axis. Similar to Figure 3 we observe a fast and significant barrier increase as more layers are added. In comparison to Figure 3 the magnitude of test barriers are shifted to lower values as the test accuracy is lower than train accuracy.

E.5 SIMILARITY OF $\mathcal{S}$ AND $\mathcal{S}'$ ON THE TEST SET

We note SA success on test barrier removal by comparing Figure 21 and Figure 22. SA performance on $\mathcal{S}$ and $\mathcal{S}'$ yields similar results for both before and after permutation scenarios. Such similar performance is observed along a wide range of width and depth for both MLP and Shallow-CNN over different datasets(MNIST, SVHN, CIFAR10, CIFAR100). We look into effects of changing

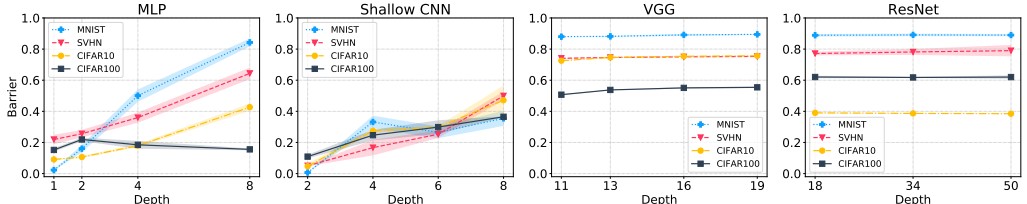

Figure 20: **Effect of depth on barrier size (Test).** From **left to right** MLP, Shallow-CNN, VGG(11,13,16,18), and ResNet(18,34,50) architectures and MNIST, CIFAR-10, SVHN, CIFAR-100 datasets. For MLP and Shallow-CNN, we fixed the layer width at $2^{10}$ while adding identical layers as shown along the x-axis. Similar behavior is observed for fully-connected and CNN family, *i.e.*, low barrier when number of layers are low while we observe a fast and significant barrier increase as more layers are added. Increasing depth leads to higher barrier values until it saturates (as seen for ResNet).

model size in terms of width and depth as in earlier Sections, and note that similar trends hold for before and after permuting solution $\theta_1$.

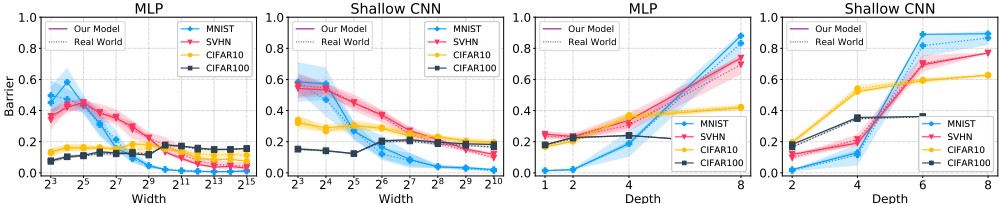

Figure 21: **Effect of width and depth on barrier consistency between real world and our model** *before* **permutation (Test).** Search space is reduced here *i.e.*, we take two SGD solutions $\theta_1$ and $\theta_2$, permute $\theta_1$ and report the barrier between permuted $\theta_1$ and $\theta_2$ as found by SA with $n = 2$. Similarity of loss barrier between real world and our model is preserved across model type and dataset choices as width and depth of the models are increased.

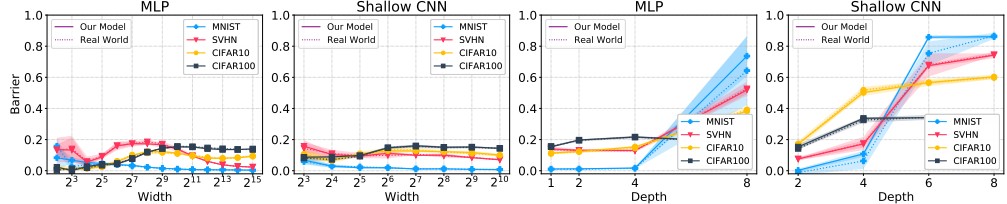

Figure 22: **Effect of width and depth on barrier consistency between real world and our model** *after* **permutation (Test).** We observe that reducing the search space makes SA more successful in finding the permutation $\{\pi\}$ to remove the barriers. Specifically, SA could indeed find permutations across different networks and datasets that when applied to $\theta_1$ result in almost zero test barrier *e.g.*, MLP across MNIST dataset where depth is 1 and width is larger than $2^6$, MLP for MNIST where depth is 2 and 4, and width is $2^{10}$, Shallow-CNN for MNIST where depth is 2, Shallow-CNN for SVHN where depth is 2 and width is $2^{10}$.

