# OpenReview forum: "The Role of Permutation Invariance in Linear Mode Connectivity of Neural Networks"
_ICLR.cc/2022/Conference — ICLR 2022 Poster_

### Official Review · Reviewer_D6tW · 2021-11-02

**Correctness:** 3
**Technical Novelty And Significance:** 2
**Empirical Novelty And Significance:** 3
**Recommendation:** 6
**Confidence:** 4

**Main Review:**

The paper makes a rigorous empirical study of their main claim across several datasets and many architectures. The barrier predictions of their model using SA are convincing.

One concern remains on the novelty of the conjecture. How do the authors compare their conjecture to the Theorem 4.2 of Simsek et. al., 2021 (https://arxiv.org/pdf/2105.12221.pdf)? It looks like this theorem has already proven the conjecture for two-layers networks for a special activation function and in an infinite data regime.

**Summary Of The Paper:**

The authors empirically study the linear mode connectivity property for four model types, their scalings, and four datasets. Their main contribution is a conjecture: 'all solutions can be linearly connected by permuting the hidden neurons of one of the solutions.'

**Summary Of The Review:**

The authors study the effect of width, depth, skip connections on the barriers (via lines, i.e., LMC) after having rotating one of the solutions with a suitable permutation.

---

> ### Author Response · Authors · 2021-11-23
> **Response to the Reviewer "D6tW"**
>
> Thanks for your valuable feedback. We appreciate you recognizing the novelty of the theoretical and empirical aspects of the work.
>
> ## Simek et al.
> We cited Simek et al. in our related work and think that our submission shares some important ideas with their work. Below we highlight the similarities and differences between these two works.
>
> First, their Theorem 4.2 basically proves our conjecture for MLPs with one hidden layer and for a specific form of activation functions. The authors provide a classification of 0-type neurons which may be useful when searching for permutations in practice and allow optimizing our SA performance on wide networks.
> Second, we also believe the differences between mildly and vastly overparameterized regimes described by Simek et. al. are related to our observations from the obtained experimental results. We observe an increase of the barrier size in our plots (Figure 2) happens in the regime when the number of saddle points >> the number of minima. In our paper, we refer to the deep double descent phenomenon to explain the peak.
> Their vastly overparameterized regime (where the number of saddle points is much lower than minimas) coincides with our over-parameterized setting (very large width) where we have a large flat region. However, they did not cover the under-parameterized regime while we do. This regime is covered in our plots before the peaks happen. Last but not least, we also looked at the effect of depth on the barrier size (Section 2.2: Depth) where we observe that for deeper networks the barrier remains high everywhere. It is worth mentioning that we carefully designed the experiments and trained networks to cover all these three regimes. Further investigation of the relationship between these phenomena is an interesting direction for future work.

---

> > ### Comment · Reviewer_D6tW · 2021-11-29
> > **Thanks for the detailed comparison**
> >
> > I thank the authors for their reply and the detailed comparison between their work and Simsek et al. There is indeed sufficient novelty in this paper for acceptance, such as the under-parameterized regime and the effect of task difficulty on the barrier size.
> >
> > Yet, as the authors explained in their rebuttal, Simsek et al. have already proven their conjecture for the case of the one hidden layer MLP. Unfortunately, this important and relevant point is not stated in the manuscript (although the paper is indeed cited).
> >
> > I invite the authors to update the manuscript to openly state that **' the conjecture has already been proven rigorously for the simplest case'**. This modification would only increase the quality of the paper and give the conjecture a more solid ground.

---

> > > ### Author Response · Authors · 2021-12-01
> > > **Response to the Reviewer "D6tW": final remarks**
> > >
> > > We would like to thank you for your review and appreciate you recognizing the novelty of our work. We will reflect your concern clearly in the final version.

---

### Official Review · Reviewer_vZXp · 2021-11-03

**Correctness:** 3
**Technical Novelty And Significance:** 3
**Empirical Novelty And Significance:** 3
**Recommendation:** 6
**Confidence:** 4

**Main Review:**

Strengths:
- The experiments are fairly extensive covering several network architectures over MNIST, Cifar10, SVHN and Cifar100.
- Though the computational burden of checking such a conjecture exhaustively is intractable, the authors use innovative approximations that might still be describing the real behavior.
- Linear Mode Connectivity is currently of great interest to the community due to the Lottery ticket hypothesis and may hold key insights into the learning dynamics of SGD on overparameterized DNNs.

Concerns/Comments:
-  In Sec 2.2, and for most of the main paper, the authors restrict their attention to training loss barrier as the "barrier to linear interpolation". I understand that this is something that can be computed at train time, but isn't the claim of linear interpolation to do with the "true" landscape of the loss function? If so, I feel understanding the test loss barrier is more interesting. I'm happy to see that it has been discussed in the appendix, but I would like to see a more thorough treatment for it.
- All the plots have a maximum barrier of $1.0$ even though I don't see any normalization in the computation of the barrier function. Is this just an artifact of the specifics of the problem? If there is no normalization, I find it hard to understand what value of barrier is large enough to be considered a refutation of linear interpolation.
- The authors note that VGG and ResNet architectures have a high barrier at all widths. Is there any explanation for why this might hold? Does it imply that SGD somehow has different dynamics on feedforward networks when compared to ResNets? And if so, how does linear mode connectivity now interact with generalization? I understand that these are very hard questions, but I thought I'd ask more out of curiosity than a criticism of the work.
- In footnote 2, it is mentioned that the experiments are conducted over $5$ pairs of solutions and that the barrier is reported at $\alpha=1/2$. Is it ensured anywhere that the solutions in those $5$ pairs are reasonably far away from each other? Reporting the barrier at $\alpha=1/2$ makes sense if the solutions were actually linearly connected, but would be unrepresentative if they were not. Something like a weighted average over the interval might make more sense to be sure of the findings.
- In section 3.1, it is stated that "In the case of permutations, all permutations are equally likely for SGD". Is there a reference for this claim?
- Section 3.3 has a typo "Bellow => Below"
- I am a little unsatisfied with the theoretical result since it has nothing to do with SGD which is the main topic of study. Moreover, the fact that it holds for all $x \in \mathbb{R}^d$ shows that this is more to do with the concentration we get from the random initialization than the network. However, I appreciate that the authors sufficiently state the limitations of this result.
- Sec 3.5 defines the "Our model" approach where the authors consider the set of all permutations as a proxy for the brute force search. While this is creative, I'm a little bit confused as to how to interpret the results. If the barriers were all non-existent up to permutations, then shouldn't the experiments on $S'$ show a lower barrier than the other experiments? I was under the impression that experiments in Figures 2 and 3 do not involve permutations.
- The footnote 3 in Section 4 seems to be missing.

**Summary Of The Paper:**

The paper makes a bold conjecture that all the solutions that SGD reaches on deep feedforward networks are linearly interpolatable if they are permuted correctly, and if the network is wide enough. They conduct fairly extensive experiments that are unable to refute this conjecture, but do not prove it with certainty. They prove a theorem that shows that such a result holds at random initialization for a fully connected 1-hidden layer ReLU.

**Summary Of The Review:**

I think the paper is a fairly thorough empirical evaluation and makes some valuable findings. However, I find it lacking for (what even the authors consider) a "bold" conjecture. I am a little confused as to the exact message of some of the experiments, and therefore am on the fence when it comes to recommending acceptance.

---

> ### Author Response · Authors · 2021-11-23
> **Response to the Reviewer "vZXp"**
>
> Thank you for your detailed feedback. Below, we first address your main concern about experimental results and then try to designate the issues in detail. We hope that you increase the score if you are satisfied with the answers to raised questions.
>
> ## Empirical Evidence
> We first would like to address the reviewer’s main concern regarding the message of the experiments. We believe our conjecture is bold because, if true, it allows considerably reducing the complexity of the loss landscape. However, empirically verifying/disproving such a computationally intensive conjecture is not easy. In our paper, we take the first steps towards this goal. First, we study the impacting factors on barrier size. Next, we look at the barrier similarity between our model and the real world, knowing that our model satisfies the conjecture (Figure 5). We then employ SA to find permutations that satisfy the conjecture. As we reduced the search space, SA was indeed able to find permutations that completely remove the barrier (Figure 6). We also analyze the barrier similarity between our model and the real world after permutation (Figure 7). To sum up, we provided experimental results over 3000 trained networks to support the conjecture.
>
> **In the following, we address all specific concerns raised by the reviewer in detail.**
>
>     Test loss
> We covered the results for the test barrier in the appendix (E4 and E5), where we observe the same patterns regarding the barrier size, supporting the conjecture. We are focused on the train barrier in the main text of the paper since looking at the test set adds another layer of complexity to the problem (=generalization), which is not the main concern of this study.
>
>     Maximum barrier of 1.0
> Thanks for your careful investigation. We are sorry that we confused the reader. To allow easy comparison between plots, all the barrier plots in the paper represent the accuracy barrier. We edited the text and pointed this out in Section 2.2, footnote 2. We consider barrier values close to zero (<2%) as linear mode connected.
>
>     Barriers VGGs and ResNets
> We kindly refer to Section 2.2, Width and Depth, where we observe the effect of depth on the barrier of deeper networks: the barrier remains high everywhere. The main difference between Shallow CNN and VGG16 is depth. We first observed a saturated barrier in deep conventional networks (VGG and ResNet) and then hypothesized that this might be due to the effect of depth. Therefore we designed a Shallow CNN architecture to study the effect of depth on the size of the barrier in convolutional networks.
>
>     ɑ=1/2
> Related works [Frankle et al, ICML 2020] showed that two randomly initialized trained solutions would end up in different basins. We are using α = ½, as a tool to test if two solutions reside in one basin, and if not we report the barrier at this point. We also tried with different alphas and would like to refer to Footnote 2: In our experiments, we observed that evaluating the barrier at α = ½ is a reasonable surrogate for taking the supremum over α (the difference is less than 10−4 ). Therefore, to save computation, we report the barrier value at α = ½.
>
>     In the case of permutations, all permutations are equally likely for SGD
> That is a consequence of the following facts: 1- All permutations of initialization are equally likely. 2- For any fixed seed, running SGD on any permutation of the initial weights is equivalent to the same permutation applied on the SGD solution from the original initial weights. We will add a simple proof to the appendix and refer to it in the final version.
>
>     Theory: SGD
> We agree that a theoretical result on SGD behavior would have been more satisfactory. However, we want to highlight that we believe SGD becomes relevant in the conjecture mostly because of its property to converge to “balanced” weights and leaving out most solutions that one can find by unit-rescaling (see Section 3.1). Moreover, we believe our theoretical result is useful both by showing the mechanism that enables the conjecture and as a basis to drive more advanced theoretical results in the NTK regime which typically has heavy reliance on the initial weights.
>
>     Sec 3.5 defines the "Our model" approach”:
> You are absolutely right that Figures 2-4 do not involve permutation, and only investigate the impacting factors over barrier size. We state our conjecture in section 3.2 and in the rest of the paper we try to illustrate supporting evidence for the conjecture. We know that S’ satisfies the conjecture. Therefore to support the conjecture, we show that S and S’ are similar in terms of barriers. Such similarity between S and S’ is first observed in Figure 5, where we look at the barrier between pairs of randomly trained solutions (S) and pairs of random permutations of solutions (S’). The same observation is also made after using simulated annealing in Figure 7.

---

> > ### Comment · Reviewer_vZXp · 2021-11-27
> > **Response to rebuttal**
> >
> > Thank you for the detailed response.
> >
> > - [Test loss]: I disagree that this is not the main concern of this study since there is an implicit assumption that the "loss landscape" is similar for both the training and test data. But I do agree that it is possibly more complicated to reason about.
> >
> > - [Maximum barrier of 1.0]: Thank you for the clarification.
> >
> > - [\alpha=1.2]: I understand that the empirical evidence pointed to this equivalence. But the argument for using it is a little circular in nature since this holds only when there is linear mode connectivity (which is what you are trying to prove). Neverthless, I understand that is impractical to evaluate the barrier at many values.
> >
> > - [SGD permutation equivalence]: I think such a statement would need smoothness/convexity assumptions on the loss function. But I'm not entirely sure. I would be happy to see the proof.
> >
> > - [Sec 3.5]: Thank you for the clarifications.
> >
> > While some of my concerns are not fully addressed, I do appreciate that this work can be considered a first step in proving a bold conjecture such as this. I also acknowledge that the experiments are quite thorough and therefore will be increasing my score.

---

> > > ### Author Response · Authors · 2021-12-01
> > > **Response to the Reviewer "vZXp": final remarks**
> > >
> > > We are very happy to see that some of your concerns are addressed. We would like to thank you for the constructive discussion and also for increasing your score. We believe that we took the first step and hope that this work would draw the community's attention to the importance of understanding neural networks loss landscape through the lens of invariances.

---

### Official Review · Reviewer_EfdQ · 2021-11-03

**Correctness:** 4
**Technical Novelty And Significance:** 3
**Empirical Novelty And Significance:** 3
**Recommendation:** 8
**Confidence:** 4

**Main Review:**

If true, this conjecture would have significant implications for both our theoretical understanding of neural networks and our use of them in practice (e.g. ensembling). The challenge of course is finding evidence for it, given that optimizing permutations between large neural networks is computationally infeasible. I think the authors take a meaningful first step in showing that a simple simulated annealing algorithm is largely unsuccessful in finding winning permutations. However I wish they had explored this direction a little further - perhaps by using other heuristics to preliminarily sort the neurons (e.g. weight norms or activations on a set of test images), and then further refining the permutation from this starting point.

As a second idea, since the function implemented by a neural network is invariant to permutation of its hidden neurons, if the authors' conjecture is correct then for two SGD solutions NN1 and NN2 there exists a solution NN2' (a permutation of NN2) which implements the same function as NN2 but lies in the same basin as NN1. Rather than finding NN2' by identifying a winning permutation, could NN2' be found by fine-tuning NN1 to implement the same function as NN2? without leaving the low-loss basin?

The authors' approach of comparing the loss barriers between different SGD solutions to the loss barriers between permutations of the same solution is interesting, and the close match is certainly intriguing. But it is only a small step towards demonstrating that their conjecture holds. I was left wondering whether there are other measurable properties of pairs of neural networks (beyond loss barriers) which could provide further evidence for (or against) their claim. For instance, some recent works [1] have shown that ensembling networks which lie in different loss basins yields significantly better performance (diverse predictions etc) than ensembling networks which lie in the same loss basin. Could a property like this be used to distinguish the real-world setting $S$ from the authors' model $S'$?

The paper is well written and clearly organized. Experiments are well-motivated and carefully described. And I think the investigations of width, depth, task and architecture on loss barrier size are independently interesting.


[1] Fort, Stanislav, Huiyi Hu, and Balaji Lakshminarayanan. "Deep ensembles: A loss landscape perspective." arXiv preprint arXiv:1912.02757 (2019).


**Summary Of The Paper:**

The authors present a bold and thought-provoking conjecture: neural networks trained with SGD converge to the same low-loss basin, up to permutations of their hidden neurons. They go on to provide some limited but intriguing evidence in support of this conjecture. First, they prove that it holds for sufficiently wide one-hidden-layer neural networks at random initialization. Second, they empirically show that the loss barriers between different SGD solutions are similar in magnitude to the loss barriers between different permutations of a single SGD solution, across a range of models, tasks, and hyperparameters (e.g. width, depth). The paper also includes a number of experiments investigating the effects of width, depth, task, and architecture on loss barrier size.

**Summary Of The Review:**

Overall, I think the conjecture presented by the authors is highly interesting and it is the first time I have seen it in writing. While the authors' empirical and theoretical results fall far short of verifying their conjecture in realistic settings, I think they are sufficient to recommend acceptance.

---

> ### Author Response · Authors · 2021-11-23
> **Response to the Reviewer "EfdQ"**
>
> Thank you for your feedback. We are grateful for the acknowledgment of the significant impact our work might have on the understanding of neural networks, in both theory and practice. If you find our contributions to be significant and your main concerns are addressed, we would appreciate it if you increase your score.
>
> ## Improve empirical results
> We absolutely agree with the reviewer that the provided empirical evidence is only the first meaningful step to “verify” the conjecture. Here we address the reviewer’s main concern about the steps to support the conjecture. Our extensive experiments over 3000 neural networks illustrate the similarity between the “real world” and “our model” in all settings. In the main text, we reduced the search space, showing that it improves the SA performance drastically. In the new rebuttal version, we added an experiment using neuron similarity to improve the search algorithm. In addition, we added another experiment using finetuning to improve the performance of ensembling.
>
> **Below we address all concerns raised by the reviewer.**
>
>
>     Heuristics to improve the search
> Thanks for your suggestion. In the last few days, we tried to improve the search algorithms by using heuristics-based neurons’ similarity. Specially we refer to [1], Equation 12, where the authors used $f(weights, post activations)$ to find pairs of neurons that could be zipped together. We applied the same method, called "Functional Difference (FD)", and observed that FD outperforms SA significantly in finding better permutations (see Figure 15 in the new rebuttal version). We edited the draft and would like to draw your attention to Section B in the appendix for more details on this experiment and the obtained results.
>
>     Finetuning to improve ensembles
>  That is indeed a nice direction to follow up. Fine-tuning NN1 to find NN2’ is nice but the challenge is that the ultimate solutions may lose functional diversity, which is a prerequisite to building high-quality ensembles. Motivated by your suggestion, we came across a recent work by Worstman et al. [2]. They start from two random initialization and try to enforce solutions in a line. As they also enforce solutions to be on a line from the start of training, we believe such a method would limit the solutions to be as far as possible in the function space. Instead, we took our Functional Difference approach (Section B in appendix) to match the solutions and then follow up with another finetuning step, similar to their work to make them literally in one basin. We specifically added Section C to add the mentioned experiment. Our preliminary results show that such a method could indeed find better solutions and outperform the learning subspace method.
>
>
>     Ensembling not on the same basin
> Stanislav et. al., investigated ensembling over predictions of two solutions, instead here we focus on averaging solutions in the weight space. One advantage of the latter is that we only need to save one model for inference. This is more important for resource-constrained settings. Secondly, a very recent study by Wortsman et. al. [3] showed that ensembling in the weight space provides better out-of-distribution generalization than ensembling in the output space.
>
>  > References
>
> [1] He, Xiaoxi, Zimu Zhou, and Lothar Thiele. "Multi-task zipping via layer-wise neuron sharing." arXiv preprint arXiv:1805.09791 (2018).
>
> [2] Wortsman, Mitchell, et al. "Learning Neural Network Subspaces." arXiv preprint arXiv:2102.10472 (2021).
>
> [3] Wortsman, Mitchell, et al. "Robust fine-tuning of zero-shot models." arXiv preprint arXiv:2109.01903 (2021).

---

> > ### Comment · Reviewer_EfdQ · 2021-11-30
> > **response to rebuttal**
> >
> > Thank you to the authors for performing these additional experiments in response to my comments.
> >
> > > Heuristics to improve the search
> >
> > I am very glad the authors followed up on this suggestion, and it is interesting to see that the simple heuristic in [1] performs substantially better in finding winning permutations than simulated annealing does. It suggests that further exploration of heuristics could reduce the loss barriers even further. However the loss barriers between "winning" permutations are still far from zero in almost all cases, and so this still doesn't really provide evidence for the authors' conjecture. And broadly speaking I'm not really sure what to make of the simulated annealing results (Figs 6,7), which indicate that simulated annealing hardly reduces the loss barriers at all from their original values, especially as compared with the new technique. These to me don't provide compelling evidence for the authors' conjecture. To me, the only real empirical evidence at the moment is the suggestive observation that loss barriers look similar between independent optima and permutations of the same optimum (Fig 1, right, Fig 2).
> >
> > > Finetuning to improve ensembles
> >
> > If I understand correctly, these new experiments provide evidence that functionally diverse solutions can be found within the same low-loss basin. While this doesn't go quite as far as what I'd proposed, nor directly support the authors' conjecture, I think it is an interesting result and I believe it strengthens the paper.
> >
> > I appreciate the authors performing these additional experiments, and I believe the results are intriguing - so I will be increasing my score.

---

> > > ### Author Response · Authors · 2021-12-01
> > > **Response to the Reviewer "EfdQ": final remarks**
> > >
> > > We would like to thank you again for your suggestions and for increasing your score.
> > > We agree to the reviewer that SA could not indeed prove/disprove the conjecture. We took the first step with SA to bring supporting evidence in the similarity of the barriers between S and S" in all settings, before and after permutation. As discussed earlier, we plan to extend the results using the new matching heuristics and will reflect the new results in the final version.
> > > The main motivation for the new experiment for the "Learning subspace" method, is not to prove the conjecture, but to show ways how to improve the ensembling method using functional difference and subspace method. We would like to thank you for this motivation.

---

### Official Review · Reviewer_iy5q · 2021-11-03

**Correctness:** 4
**Technical Novelty And Significance:** 3
**Empirical Novelty And Significance:** 4
**Recommendation:** 8
**Confidence:** 3

**Details Of Ethics Concerns:**

None.

**Main Review:**

## Strength

- Understanding properties about the optimization landscape of neural networks is a central problem in deep learning. While there has been lots of theoretical studies, they typically apply only to particular styles of data models or rely on particular assumptions on the initialization. This paper instead directly studies properties of the optimization landscape through large-scale experiments. The findings are both interesting and novel.

- The authors have stated a Conjecture that could potentially inspire future studies. This Conjecture is in line with existing results in the literature. A preliminary result in the large-width limit is provided to support the Conjecture.

- The scale of the experiments is on a large range of datasets and models, with over 3,000 runs, giving validity to the supporting arguments.

## Weakness

- Based on Conjecture 1, one would expect the barrier to go to zero, provided with sufficient computation to find the “right” permutation to apply to the weight matrices. Figure 6 goes into supporting this Conjecture, by showing that Simulated Annealing helps reduce the barrier. However, the reduction in the barrier is marginal at best. Thus, I am not entirely convinced by the result of this experiment.
    - Could the authors show a scaling cost as for whether the barrier would continue to decrease, provided more computation? For example, suppose the barrier continues to decrease as the amount of computation increases and does not plateau, then this would suggest that with enough computation, the barrier could eventually go to zero. The current result shown in Figure 6 looks marginally supportive at best.
- Based on the nature of this paper, I think the discussions should be stated more concretely.
    - For example, the authors state “Another area where our analysis is of importance is for ensembles and distributed training.” It’s unclear what this refers to.
    - It’s also unclear when the authors state “whether there is a one-to-one mapping between lottery tickets and permutations. We believe our analysis laid the ground for investigating these important questions…”

**Questions**
- From the paper by Zhang et al. (ICLR’17), we know that neural nets can memorize random labels and regularization methods can help mitigate such memorization (Figure 2a) though not entirely. Would Conjecture 1 still be true if one takes into account the effect of regularization? For example, while the permutation invariance property holds for regularization methods such as weight decay, it does not hold for $\ell_1$ penalty or data augmentation.
- Additionally, it would be interesting to see whether the barrier changes if one starts to inject random labels into the training dataset, similar to Zhang et al.’s experiments.
- I was also curious what the prediction performance of the linearly-interpolated model is. For example, it would be interesting to see a figure similar to Figure 2 but with the y-axis being the prediction accuracy.

**References**
- Understanding Deep Learning Requires Rethinking Generalization. Zhang, Bengio, Hardt, Recht, and Vinyals. ICLR’17.

**Typos**
- Extra space on page 2: “In Section 4”
- Missing white space on page 8: “$\Psi$ to”

**Summary Of The Paper:**

This paper studies the optimization landscape of neural networks. In particular, this paper touches on the question of whether two local minima of the optimization landscape are connected by a “line” of linearly-interpolated neural networks. To probe this question, this paper evaluates the loss gap (so called “barrier”) between two local minima and their linear interpolation. The finding is that this barrier is non-zero. Next, the authors conjecture that if one takes into account permutation invariance of a local minimum, the barrier should be reduced to zero. To support this conjecture, the authors considered a simulated annealing algorithm to search for permutations of the weight matrices, to show that the barrier reduces after applying the simulated annealing algorithm

**Summary Of The Review:**

This paper presents several interesting and novel ideas regarding the linear mode connectivity of the optimization landscape of neural networks. The findings are supported by extensive experiments. An interesting Conjecture is stated, which could be interesting for future studies. On the other hand, I think the writing of this paper could still be improved.

**Update after rebuttal:** The rebuttal addressed my main concerns mentioned under the Main Review section; thus, I increased my score. I particularly appreciated the empirical strength and the insights of this paper.

On the other hand, I agree with other reviewers that the authors need to clearly state the version of the conjecture that is already proved in prior works.

My other remaining concern is about the loss function used in the proposed conjecture. As in the current form, the conjecture is stated for the minimizers of generic loss landscapes of neural networks. However, it doesn't explicitly clarify whether there are any regularization penalties (e.g., $\ell_2$ or $\ell_1$ loss of weight matrices) in the loss function, which I believe is used in the experiments (e.g., weight decay). I think this aspect needs to be stated more carefully.

---

> ### Author Response · Authors · 2021-11-23
> **Response to the Reviewer "iy5q"**
>
> Thanks for your valuable feedback. We are grateful for recognizing the thoroughness of our theory and experiments. We sincerely hope that you consider increasing your score if your main concern is addressed.
>
> ## Our approach to the conjecture:
> Here we address the reviewer’s main concern regarding empirical steps on search algorithm performance. We started by permuting all solutions to one large flat basin (Figure 6, left). Similar to the main concern raised by the reviewer, we also noticed that SA is neither able to make the barrier zero nor to disprove the conjecture. We raised this specific question: Is this due to the SA performance in a very large search space or that is an observation that refutes the conjecture? We proposed two empirical approaches to answer this question: namely, (1) the search space reduction, and (2) designing “our model” which provides a good approximation for all solutions in the “real world”. Our results on reducing the search space (Figure 6, right) show that SA performance could indeed be improved. However, the barrier remains similar in both “real world” and “our model”, supporting the conjecture. We also added the results of the scaling experiment, suggested by the reviewer, confirming that the barrier would continue to decrease, providing more computation to SA.
>
> **Below we address all specific questions raised by the reviewer.**
>
>
>
>     SA performance
>
>  We would like to emphasize that our claim is not on the performance of SA for the combinatorial search in such a large search space. Instead, our experimental design lies on two principles to support the conjecture: (1) “Our model” satisfies the conjecture. (2) We showed that “our model” and “real world” exhibit the same barrier behavior across all settings. SA was able to reduce the barrier by half for MNIST and one-layer MLP where there are 128! permutations and also find the best permutation in multiple settings. Reducing the search space could indeed help SA to find better permutations (Figure 6: Right).
>
>     Scaling experiment
>
> We edited Section A.4 in the Appendix to include another experiment to address the reviewer's concerns. Figure 14 shows that increasing the number of steps exponentially helps SA to find better solutions. Table 2 (Appendix A.4) shows that as the number of steps increases ($10\times$), $\Delta$ moves towards $2$ i.e. 50\% reduction in barrier ($\Delta>0$ means barrier does not plateau). However, running SA for 50K steps takes 10K seconds on an n1-standard-8 GCP machine (8 vCPU, 30 GB RAM) with 1xV100 GPU. Due to limited computational resources, we set the number of steps to 50K.
>
>     ensembles, distributed training, and LTH
>
> Thanks to your comment we edited the text to include the discussion of model ensembles and distributed training and also the connection to LTH. We specifically refer to the third paragraph in the discussions and conclusions in the new rebuttal version.
>
>     Generalization of the conjecture
>
> This is indeed a nice question. We know that all mentioned ablations (regularization, data augmentation, etc.) would not affect a permutation, e.g. flipping an image or changing the contrast do not have any effect on permuting neurons in MLP or out-channels in convolutional networks. However, more interesting is whether these parameters have any effect on the conjecture or not? Our experiments show that barrier behaviors in the case of explicit regularization, e.g. weight decay is the same as SGD with implicit regularization. In our experiments, we also used data augmentations (Appendix A1). We believe the conjecture could be extended to the most common deep learning optimizers given their similarity.
>
>
>     Injecting random labels
>
> We added a new section in the appendix (E1 in the new rebuttal version) to address the reviewer’s interesting question. Our results over 5 different runs (Figure 15: right) show that the barrier size behavior does not change. i.e. similar to deep double descent, the barrier first increases and then decreases. The peak in barrier size coincides with the peak in the loss.
>
>     Accuracy at the barrier
>
> We are sorry that we confused the reader. To simplify the comparison between plots, all the barrier plots in the paper show the accuracy barrier. We edited the text and pointed this out in Section 2.2, footnote 2.
>
> > References:
>
> [1] Frankle, Jonathan, et al. "Linear mode connectivity and the lottery ticket hypothesis." International Conference on Machine Learning. PMLR, 2020.
>
> [2] Fort, Stanislav, Huiyi Hu, and Balaji Lakshminarayanan. "Deep ensembles: A loss landscape perspective." arXiv preprint arXiv:1912.02757 (2019).
>
> [3] Izmailov, Pavel, et al. "Averaging weights leads to wider optima and better generalization." arXiv preprint arXiv:1803.05407 (2018).
>
> [4] Wen, Yeming, Dustin Tran, and Jimmy Ba. "Batchensemble: an alternative approach to efficient ensemble and lifelong learning." arXiv preprint arXiv:2002.06715 (2020).

---

> ### Author Response · Authors · 2021-12-01
> **Response to the Reviewer "iy5q": Final remarks**
>
> We are very happy that your concerns are addressed in the rebuttal version and hence would like to thank you for increasing your score.
> Sure. We would clearly cite the related works in relation to the conjecture in the final version.  Regarding the regularization penalties, no we have not used any regularization in our experiment (See A.1 for training hyperparameters). We rely on the implicit regularization of SGD.

---

### Public Comment · ~Sidak_Pal_Singh1 · 2021-11-15
**Blatant ignorance of closely related work and questions over 'novel' contributions**

Hi,

This work considers an interesting direction and presents interesting analyses, yet the underlying fact is that there is a significant ignorance of a closely related work, which raises important questions of the *actual novelty of the presented results.* In particular, I am talking about a paper of mine on 'Model Fusion via Optimal Transport', available since 2019 at https://arxiv.org/abs/1910.05653 and which featured at NeurIPS.  Let me elaborate:

We considered the problem of fusing two or more trained neural networks, where of course it is important to take into account the permutation symmetries. Thus, we presented an algorithm based on Optimal transport which does a layer-wise alignment and then averages the networks.  In the course of this, *we provided many empirical analyses which would already establish **clear antecedents** of the several results discussed in the authors' work.* The overarching illustration for permutation symmetries significantly reducing barrier size is presented in Figure 2, but to specifically and concretely list some of these antecedents:

1. *Effect of width on barrier size:* We show in Table S11 that the barrier sizes  [a] generally decrease with increasing width. In fact, from the same table, we can see that even the barrier for vanilla average of the network parameters (which would correspond to linear interpolation by ignoring permutation symmetries) decreases. Importantly, this implies that effect of permutation symmetries itself is decreasing --- this brings how significant is the conjecture then!


2. *Effect of different architectures (Deep convolutional and residual networks), datasets  on barrier size*: Our results clearly indicate, see Table 1 and Fig 2, that VGG, ResNet models have larger barriers than that obtained in the case of MLP.  Similarly, we find large barriers for tasks of increasing difficulty: MNIST, CIFAR10, CIFAR100.



3. *Repeated (ignorant) statements are made about model ensembling*, e.g., "Note, our conjecture also has great practical implications for model ensembling and parallelism since one can average models that are in the same basin in the loss landscape", "Another area where our analysis is of importance is for ensembles and distributed training. If we can track the optimal permutation (that brings all solutions to one basin), it is possible to use it to do weight averaging and build ensembles more efficiently.".  ***Our work has precisely has established this aspect, in practice, for multiple deep neural networks**!*


4. *Algorithm for winning permutation:* Not only we have established 'practical implications' of the conjecture, but that too with the help of an efficient algorithm -- and not by a mere brute-forced search over permutation maps! [b]


5. *The conjecture itself:* If the merit of conjecture (as evaluated in your work) is by only showing that the barrier in case of large width fully-connected networks -- after accounting for permutation symmetries -- is small or none, then our work already is providing *initial* evidence into this (see point 1).



6. *Debate about the proposal of conjecture in the first place:* Disclaimer, this is perhaps a subjective viewpoint of mine. The reason is that there seems to be a non-zero barrier even after applying the search (or in our work, the OT fusion algorithm) for both not-extremely-wide but shallow networks (let alone deeper networks or even CNNs/ResNets). It is a valid point that it is hard to judge whether there is due to algorithmic reasons or otherwise. But, still even in the case of (just) small width (2^4) networks the search fails to find a zero barrier --- despite the search space can be completely brute-forced as stated in the paper. Now, of course, you delimit this by asking for large enough width in your conjecture --- but then, I'd wonder, how significant is this conjecture (see point 1 about vanilla averaging itself being not too bad for large width networks)?


a: There are some minor differences: our definition of barrier is based on test error, and mostly consider alpha = 0.5.
b: Note, this is done via exact OT, and we do uncover proper permutation matrices.

**Final Remarks:** Where credit is due, this work does a nice job at *further* exploration of this role of permutation symmetries in more settings. Nevertheless, my critique centers on the way this paper articulates its findings with a complete disregard of prior work, and importantly, where most of the ideas and contributions already have their roots and origins.

Sidak

---

> ### Author Response · Authors · 2021-11-23
> **Response to Sidak**
>
> Dear Sidak,
>
> Thank you for your valuable feedback and for bringing your work to our attention. While we cite similar works to yours (He et al 2018, Brea et al 2019, Tatro et al. 2020), we were not aware of your paper up until now and will cite it in the final version.
>
> Despite similarities between our works mentioned in your comment, we would like to note some substantial differences. Our contributions stand on two main aspects:
>
> Firstly, we looked at the effects of width, depth, and task complexity over barrier size along with different datasets and architectures. These effects were studied under three different regimes, namely under parametrized (before the peak), overparameterized (where peak happens), and vastly overparameterized (after the peak). These experiments were carefully designed to cover different datasets (MNIST, CIFAR10, CIFAR100, ImageNet) and architectures (MLPs with multiple hidden layers, convolutional networks with different ranges of layers, and also residual networks) and ran multiple times to support the findings. For example, as we increase the width, we observe an increase and then decrease of the barrier, which is similar to the deep double descent phenomenon in the loss domain. Based on our extensive experiments we could confirm that the peak for barrier size matches the peak of loss in deep double descent. Our extensive experiments also made us able to give intuitions about the role of depth in barrier increase in modern network architectures.
>
> Secondly, the main focus of the paper is not to propose a search algorithm (or a matching algorithm), but the conjecture is about the shape of the loss landscape and the structure of basins. Our conjecture is not concerned about the exact permutation between two networks, but rather hypothesizes the existence of “a” permutation that brings two (or more) solutions in the same basin, hence making them linearly mode connected.
>
> It is also worth mentioning that, motivated by one of the reviewer’s concerns, we worked on an improved version of our search algorithm based on Functional Difference [He et al, Nurips 2018]. The authors define Functional Difference as a product of weights and post-activations (Equation 12), very similar to “Alignment strategies” in your paper. Our preliminary results show that Functional Difference finds better permutations. We will use your method for neuron matching and compare it with the functional difference for the final version.

---

### Decision · Program_Chairs · 2022-01-20

**Decision:**

Accept (Poster)

**Comment:**

This paper investigates the linear mode connectivity of the loss landscape of neural networks, i.e. whether a convex combination of two parameters of local optima on the SGD paths has low loss values (i.e. low barrier) up to some permutations. To probe this question, this paper empirically studies the loss gap, named as “barrier”, between two local minima and their convex combinations or linear interpolation. Before permutations, such barriers are typically non-zero; yet, after taking into account of permutation invariance of models, such barriers could be reduced along to zero with the width increasing, a main conjecture formulated in the paper. To support this conjecture, the authors proposed a simulated annealing algorithm to search for such permutations, demonstrating that the barrier reduces after such permutations.

The reviewers unanimously accept the paper, if the authors make the proposed improvement in the final version. In particular, a reader points out a paper by Singh and Jaggi, 'Model Fusion via Optimal Transport', NeurIPS 2020, that supports the same conjecture with a constructive algorithm to find optimal permutations or matching using optimal transport. This should be included in the final version as the authors replied.